# Your Absorbing Discrete Diffusion Secretly Models the Conditional Distributions of Clean Data

## Abstract

Discrete diffusion models with absorbing processes have shown promise in language modeling. The key quantities to be estimated are the ratios between the marginal probabilities of two transitive states at all timesteps, called the concrete score. In this paper, we reveal that the concrete score in absorbing diffusion can be expressed as conditional probabilities of clean data, multiplied by a time-dependent scalar in an analytic form. Motivated by the finding, we propose reparameterized absorbing discrete diffusion (RADD), a dedicated diffusion model that characterizes the time-independent conditional probabilities. Besides its simplicity, RADD can reduce the number of function evaluations (NFEs) by caching the output of the time-independent network when the noisy sample remains unchanged in a sampling interval. Empirically, RADD is up to 3.5 times faster while consistently achieving a better performance than the strongest baseline. Built upon the new factorization of the concrete score, we further prove a surprising result that the exact likelihood of absorbing diffusion can be rewritten to a simple form (named denoise cross-entropy) and then estimated efficiently by the Monte Carlo method. The resulting approach also applies to the original parameterization of the concrete score. It significantly advances the state-of-the-art discrete diffusion on 5 zero-shot language modeling benchmarks (measured by perplexity) at the GPT-2 scale.

## 1 Introduction

Auto-regressive models [1, 2, 3] have dominated the area of language modeling for many years. In particular, such models significantly benefit from large-scale transformers [4] and training data and have achieved remarkable progress [5, 6, 7, 8]. From a probabilistic perspective, the sequential sampling process of auto-regressive models is inefficient and limits the reasoning ability in nonsequential orders [9, 10]. Intrinsically, this is because such models characterize the joint distribution by the chain rule of probability, motivating research on developing other types of generative models for text.

Diffusion models [11, 12, 13] generate data in a coarse-to-fine manner efficiently [14, 15, 16, 17, 18] and all dimensions simultaneously, providing an appealing alternative to auto-regressive models. Among other efforts [19, 20, 21, 22, 23, 24, 20, 25, 26, 27, 28, 29] (see Section 5 for a comprehensive discussion), score entropy discrete diffusion (SEDD) [29] has shown promise in text generation. In particular, SEDD has achieved comparable results to auto-regressive models on 5 zero-shot language modeling benchmarks at the GPT-2 scale. Meanwhile, SEDD can reduce the number of function evaluations (NFEs) in sampling and fulfill text conditioned on prompts at different positions.

Technically, SEDD employs a discrete-state (absorbing) Markov process that adds noises to data by randomly replacing a token with a mask token [M] and then learns a reverse process to denoise from an entirely masked sentence. The key quantities to be estimated in SEDD are the ratios between the marginal probabilities of two transitive states at all timesteps, called the **concrete score**. SEDD also

proposes a "scaling trick" (see details in Section 3) that scales the output of the score estimation by a factor. The trick has been proven very effective in practice yet not fully understood in theory [29].

One of our main contributions is to reveal that the concrete score in absorbing diffusion can be expressed as conditional probabilities of clean data, multiplied by a time-dependent scalar in an analytic form. Our finding theoretically explains the benefits of the scaling trick as a reparameterization for better optimization. Motivated by the finding, we propose reparameterized absorbing discrete diffusion (RADD), a dedicated diffusion model that characterizes the time-independent conditional probabilities by removing the time embedding from the score estimation in SEDD. Besides its simplicity, RADD can significantly reduce the NFEs by caching the output of the time-independent network when the noisy sample remains unchanged in a sampling interval (see Fig. 1).

Built upon the new factorization of the concrete score, we further prove a surprising result that the exact likelihood of absorbing diffusion can be rewritten to a simple form (named denoise cross-entropy, DCE) and then estimated efficiently by the Monte Carlo method. To establish the theory, we apply a change of variable from the time $t$ to the probability that a single-dimensional token is masked at time $t$ in the forward process. By integrating the probability variable analytically, we show that DCE enumerates all orders to decompose the joint distribution auto-regressively and accumulates log densities of all conditional distributions in every order, finishing the proof. Such theoretical findings enable exact likelihood evaluation and optimization for both the original parameterization of absorbing diffusion [29] and the proposed RADD.

Empirically, RADD is up to 3.5 times faster while consistently achieving a better performance than the strongest baseline, i.e. SEDD with the scaling trick [29]. Further, the DCE loss applies to both RADD and SEDD for precise likelihood evaluation. It significantly advances the state-of-the-art discrete diffusion (i.e. SEDD [29]) on 5 zero-shot language modeling benchmarks (measured by perplexity) at the GPT-2 scale. The empirical evidence validates our theoretical findings.

In summary, this paper has several contributions:

- **Deeper understanding of discrete diffusion**: Both the factorization form of the concrete score and DCE loss for the exact likelihood computation reveal important yet overlooked theoretical properties of absorbing discrete diffusion, which explain the mysterious scaling trick, provide practice guidance, and may inspire future work.

- **Simplification**: By removing the time conditions, we reparameterize the model to focus on a time-independent conditional probability, simplifying the existing model.

- **Efficient sampling**: Leveraging the reparameterized form, RADD with a caching strategy is consistently faster while achieving a better performance than the strongest competitor.

- **Improved likelihood evaluation**: The exact likelihood evaluation approach significantly advances the state-of-the-art discrete diffusion on 5 zero-shot language modeling benchmarks (measured by perplexity) at the GPT-2 scale.

## 2    Background

In this section, we present preliminaries on continuous-time discrete diffusion models. We start with the one-dimensional case in Section 2.1, followed by the multi-dimensional case in Section 2.2.

### 2.1    Single dimension

Let $x$ denote a single dimensional sample with possible values in $\{1, \ldots, N\}$. A continuous-time discrete Markov chain at time $t$ is characterized by a transition rate matrix $\boldsymbol{Q}_t$ as follows

$$p_{t+\Delta t|t}(\hat{x}|x) = \begin{cases} \boldsymbol{Q}_t(x, \hat{x})\Delta t + o(\Delta t), & \hat{x} \neq x, \\ 1 + \boldsymbol{Q}_t(x, x)\Delta t + o(\Delta t), & \hat{x} = x, \end{cases} \tag{2.1}$$

where $\boldsymbol{Q}_t(x, \hat{x})$ is the $(x, \hat{x})$ element of transition rate matrix $\boldsymbol{Q}_t$, denoting the transition rate from state $x$ to state $\hat{x}$ at time $t$. Equivalently, we can directly define $\boldsymbol{Q}_t(x, \hat{x})$ as

$$\boldsymbol{Q}_t(x, \hat{x}) = \begin{cases} \lim_{\Delta t \to 0} \frac{p_{t+\Delta t|t}(\hat{x}|x)}{\Delta t}, & \hat{x} \neq x, \\ \lim_{\Delta t \to 0} \frac{p_{t+\Delta t|t}(x|x)-1}{\Delta t}, & \hat{x} = x. \end{cases} \tag{2.2}$$

Given the above definition, denote $\boldsymbol{P}_{s \to t}(x, \hat{x}) := p_{t|s}(\hat{x}|x)$. The following Kolmogorov's forward equation holds [26, 30]:

$$\frac{d}{dt} \boldsymbol{P}_{s \to t} = \boldsymbol{P}_{s \to t} \boldsymbol{Q}_t. \tag{2.3}$$

In practice [26, 29], $\boldsymbol{Q}_t$ is parameterized as $\sigma(t)\boldsymbol{Q}$, where $\sigma(t)$ is a scalar function and $\boldsymbol{Q}$ is a constant matrix. In this case, the solution to Eq. (2.3) can be solved analytically as $\boldsymbol{P}_{s \to t} = \exp\left((\bar{\sigma}(t) - \bar{\sigma}(s))\boldsymbol{Q}\right)$, where $\bar{\sigma}(t) = \int_0^t \sigma(s)ds$ and $\exp$ is the matrix exponential. Therefore, we can directly sample $\boldsymbol{x}_t$ from $\boldsymbol{x}_s$ in one step for any $t > s$.

Further, $\boldsymbol{Q}$ is often designed to diffuse towards a uniform distribution or an absorbing state [M]. Recent work [20, 26] suggests that the absorbing matrix achieves better empirical performance. Besides, as detailed in Section 3, the specific structure of the absorbing matrix can be leveraged to improve performance and accelerate sampling. Therefore, we focus on the absorbing matrix as follows:

$$\boldsymbol{Q}^{\text{absorb}} = \begin{bmatrix} -1 & 0 & \cdots & 0 & 1 \\ 0 & -1 & \cdots & 0 & 1 \\ \vdots & \vdots & \ddots & \vdots & \vdots \\ 0 & 0 & \cdots & -1 & 1 \\ 0 & 0 & \cdots & 0 & 0 \end{bmatrix}. \tag{2.4}$$

The time reversal of the forward process is characterized by a reverse transition rate matrix $\tilde{\boldsymbol{Q}}_t$ [31, 32], whose element from state $x$ to state $\hat{x}$ is given by

$$\tilde{\boldsymbol{Q}}_t(x, \hat{x}) = \begin{cases} \frac{p_t(\hat{x})}{p_t(x)} \boldsymbol{Q}_t(\hat{x}, x), & \hat{x} \neq x, \\ -\sum_{k \neq x} \tilde{\boldsymbol{Q}}_t(x, k), & \hat{x} = x. \end{cases} \tag{2.5}$$

Simulating the reverse process requires to learn the reverse transition rate $\tilde{\boldsymbol{Q}}_t(x, \hat{x})$. As $\boldsymbol{Q}_t(x_t, \hat{x}_t)$ is known, it is sufficient to estimate the concrete score $\frac{p_t(\hat{x}_t)}{p_t(x_t)}$ by a score network $s_\theta(x_t, t) \approx [\frac{p_t(\hat{x}_t)}{p_t(x_t)}]_{\hat{x}_t \in \mathcal{X}}$ [28]. Denoising score entropy (DSE) [29] is an effective objective to train the score network

$$\int_0^T \mathbb{E}_{\tilde{x} \sim p_{t|0}(\cdot|x_0)} \sum_{y \neq \tilde{x}} \boldsymbol{Q}_t\left(\tilde{x}, y\right) \left(s_\theta\left(\tilde{x}, t\right)_y - \frac{p_{t|0}\left(y \mid x_0\right)}{p_{t|0}\left(\tilde{x} \mid x_0\right)} \log s_\theta\left(\tilde{x}, t\right)_y + K\left(\frac{p_{t|0}\left(y \mid x_0\right)}{p_{t|0}\left(\tilde{x} \mid x_0\right)}\right)\right) dt, \tag{2.6}$$

where $K(a) := a \log a - a$. In particular, the DSE loss in Eq. (2.6) is an evidence lower bound (ELBO) of the negative log-likelihood with an unknown gap. Nevertheless, existing work [29] still employs it for training and likelihood evaluation.

After training, sampling from the model can be understood as discretizing the following process

$$\frac{d}{dt} \boldsymbol{P}_{s \to t} = \boldsymbol{P}_{s \to t} \tilde{\boldsymbol{Q}}_t, \tag{2.7}$$

where $dt$ is an infinitesimal negative timestep and the concrete score is replaced by the score network. Existing samplers include the Euler method, Gillespie method, and Tweedie $\tau$-leaping, as detailed in Appendix D.

## 2.2 Multi-dimension

The multi-dimensional cases consider a state space of size $d$ like $\mathcal{X}^d = \{1, \ldots, n\}^d$. We denote the sample as a sequence of one-dimensional data, i.e. $\boldsymbol{x} = x^1 \ldots x^d$. The transition matrix $\boldsymbol{Q}_t \in \mathbb{R}^{n^d \times n^d}$ has an exponential number of possible states, making it expensive to reverse. To alleviate this issue, existing work [26, 29] assumes independence between dimensions and each dimension is a one-dimensional diffusion process with the same transition rate matrix $\boldsymbol{Q}_t^{\text{tok}} \in \mathbb{R}^{n \times n}$.

Under the independent assumption, $\boldsymbol{Q}_t$ assigns zero values [26, 29] for all sequences with a Hamming distance larger than 1. According to Eq. (2.4), it is sufficient to model the concrete score between

sequences that differ by a Hamming distance of 1, such as $\hat{\boldsymbol{x}}_t = x_t^1 \ldots \widehat{x}_t^i \ldots x_t^d$ given $\boldsymbol{x}_t = x_t^1 \cdots x_t^d$. Therefore, the score network $\boldsymbol{s}_\theta(\cdot, t) : \{1, \ldots, n\}^d \to \mathbb{R}^{d \times n}$ is defined as

$$\boldsymbol{s}_\theta\left(\boldsymbol{x}_t, t\right)_{\hat{\boldsymbol{x}}_t} = \boldsymbol{s}_\theta\left(x_t^1 \ldots x_t^i \ldots x_t^d, t\right)[i, \widehat{x}_t^i] \approx \frac{p_t\left(x_t^1 \ldots \widehat{x}_t^i \ldots x_t^d\right)}{p_t\left(x_t^1 \ldots x_t^i \ldots x_t^d\right)},$$

which leads to the following expression to estimate the reverse transition rate matrix $\tilde{\boldsymbol{Q}}_t$:

$$\tilde{\boldsymbol{Q}}_t\left(x_t^1 \ldots x_t^i \ldots x_t^d, x_t^1 \ldots \widehat{x}_t^i \ldots x_t^d\right) = \boldsymbol{Q}_t^{\text{tok}}\left(\widehat{x}_t^i, x_t^i\right) \frac{p_t\left(x_t^1 \ldots \widehat{x}_t^i \ldots x_t^d\right)}{p_t\left(x_t^1 \ldots x_t^i \ldots x_t^d\right)} \tag{2.8}$$

$$\approx \boldsymbol{Q}_t^{\text{tok}}\left(\widehat{x}_t^i, x_t^i\right) \boldsymbol{s}_\theta\left(x_t^1 \ldots x_t^i \ldots x_t^d, t\right)[i, \widehat{x}_t^i]. \tag{2.9}$$

Existing samplers assume that each dimension is independent within a small interval $\Delta t$ and update each dimension in parallel for efficiency [29, 26].

## 3 Reparameterized absorbing discrete diffusion

In Section 3.1, we reveal that the concrete score of absorbing discrete diffusion can be reparameterized as conditional distributions of clean data, which enables efficient sampling by caching the output of time-independent network (see Section 3.2) and exact likelihood computation (see Section 3.3) by applying the change of variable from time to the probability of being masked in a single dimension.

### 3.1 Parameterizing the concrete score as conditional distributions of clean data

A key observation is that only the transition from the masked token to an unmasked token is valid in the reverse process of an absorbing discrete diffusion. In particular, according to the definition of the transition matrix of the absorbing process (see Eq. (2.4)), we have $\boldsymbol{Q}^{\text{absorb}}(\hat{x}_t^i, x_t^i) = 0$ for any unmasked $x_t^i \neq [\mathbf{M}]$ and $\hat{x}_t^i \neq x_t^i$. Therefore, the corresponding element in the transition matrix of the reverse process $\tilde{\boldsymbol{Q}}_t$ (see Eq. (2.5)) equals zero. Namely,

$$\tilde{\boldsymbol{Q}}_t\left(x_t^1 \ldots x_t^i \ldots x_t^d, x_t^1 \ldots \widehat{x}_t^i \ldots x_t^d\right) = \sigma(t)\boldsymbol{Q}^{\text{absorb}}\left(\widehat{x}_t^i, x_t^i\right) \frac{p_t\left(x_t^1 \ldots \widehat{x}_t^i \ldots x_t^d\right)}{p_t\left(x_t^1 \ldots x_t^i \ldots x_t^d\right)} = 0, \tag{3.1}$$

for any unmasked state $x_t^i \neq [\mathbf{M}]$ and $\hat{x}_t^i \neq x_t^i$ and it is unnecessary to model the corresponding concrete score $\frac{p_t\left(x_t^1 \ldots \widehat{x}_t^i \ldots x_t^d\right)}{p_t\left(x_t^1 \ldots x_t^i \ldots x_t^d\right)}$. Also, note that the concrete score always takes the value of one if $\hat{x}_t^i = x_t^i$. Therefore, we only need to characterize the concrete score for $x_t^i = [\mathbf{M}]$ and $\hat{x}_t^i \neq [\mathbf{M}]$.

Interestingly, in this case, we discover that the concrete score has a simple analytic form w.r.t. to the conditional distributions of clean data, as summarized in the following Theorem 1.

**Theorem 1.** *(Analytic concrete score in absorbing case, proof in Appendix B) For $\boldsymbol{x}_t = x_t^1 \ldots x_t^i \ldots x_t^d$ and $\hat{\boldsymbol{x}}_t = x_t^1 \ldots \widehat{x}_t^i \ldots x_t^d$, if $x_t^i = [\boldsymbol{M}]$ and $\hat{x}_t^i \neq [\boldsymbol{M}]$, the concrete score at time $t$ can be expressed as a* time-independent *conditional distribution at time zero multiplied by an analytic* time-dependent *term:*

$$\frac{p_t\left(x_t^1 \ldots \widehat{x}_t^i \ldots x_t^d\right)}{p_t\left(x_t^1 \ldots x_t^i \ldots x_t^d\right)} = \frac{e^{-\bar{\sigma}(t)}}{1 - e^{-\bar{\sigma}(t)}} p_0(\hat{x}_t^i | \boldsymbol{x}_t^{UM}),$$

*where $\boldsymbol{x}_t^{UM}$ is the vector consists of all unmasked tokens of $\boldsymbol{x}_t$.*

One immediate implication of Theorem 1 is to theoretically explain the benefit of the "scaling trick" in existing work [29] (see Appendix C.2 therein), which significantly improves the practical performance of discrete diffusion (see Table 2) but has not been fully understood.

In particular, the scaling trick divides the output of the score network $\boldsymbol{s}_\theta$ by a factor of $e^{\bar{\sigma}(t)} - 1$. Equivalently, it reparameterizes $\boldsymbol{s}_\theta(\boldsymbol{x}_t, t)$ as:

$$\boldsymbol{s}_\theta(\boldsymbol{x}_t, t) = \frac{1}{e^{\bar{\sigma}(t)} - 1} \tilde{\boldsymbol{s}}_\theta(\boldsymbol{x}_t, t) = \frac{e^{-\bar{\sigma}(t)}}{1 - e^{-\bar{\sigma}(t)}} \tilde{\boldsymbol{s}}_\theta(\boldsymbol{x}_t, t),$$

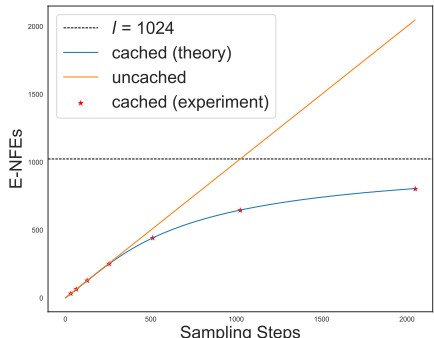

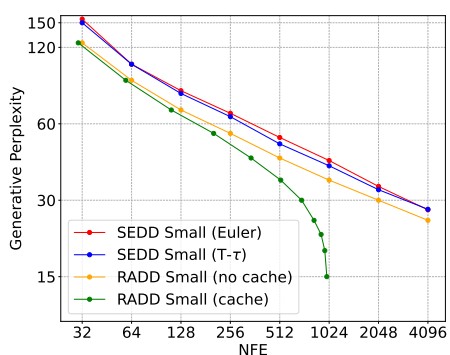

Figure 1: **Expected number of function evaluations (E-NFE) over a different number of sampling steps.** E-NFE is measured by Tweedie $\tau$-leaping method with log-linear noise schedule.

Figure 2: **Sample quality measured by perplexity ($\downarrow$).** We compare SEDD with Euler and Tweedie $\tau$-leaping (abbr. T-$\tau$) samplers, and RADD with Euler sampler. We show E-NFE for RADD with caching and NEF otherwise.

where the scaling factor coincides with the time-dependent term in Theorem 1. In the original parameterization, the score network $s_\theta$ must model the whole time-dependent concrete score. In contrast, with the scaling trick, the reparameterized score $\tilde{s}_\theta(\boldsymbol{x}_t, t)$ can focus on capturing the clean data distribution $p_0(\hat{x}^i|\boldsymbol{x}_t^{\mathrm{UM}})$ and simplifies learning, according to Theorem 1.

Further, Theorem 1 suggests that it is unnecessary to incorporate the time $t$ in the reparameterized score, and the reparameterized score $\tilde{s}_\theta(\boldsymbol{x}_t, t)$ should output a valid probability distribution. Motivated by the insights, we propose reparameterized absorbing discrete diffusion (RADD), which employs a network $\boldsymbol{c}_\theta(\boldsymbol{x}_t)$ that removes the time condition from the input and takes the softmax as final nonlinearity. Formally, we can write our reparameterization as:

$$\boldsymbol{s}_\theta(\boldsymbol{x}_t, t) = \frac{e^{-\bar{\sigma}(t)}}{1 - e^{-\bar{\sigma}(t)}} \boldsymbol{c}_\theta(\boldsymbol{x}_t). \tag{3.2}$$

In practice, we make a minimal modification of the score network in SEDD [29] for simplicity and fairness, detailed in Appendix F.1.

Moreover, RADD also enjoys a more efficient sampling process than SEDD [29] (with or without the scaling trick) based on its simplified parameterization, as presented below.

### 3.2 Efficient samplers to reduce NFE by caching $c_\theta(x_t)$

For the reverse process of an absorbing discrete diffusion, once a token is generated from [M] to an unmasked token, it never transits to another token. Therefore, for a sequence $\boldsymbol{x}_t$ of length $d$, $\boldsymbol{x}_t$ changes at most $d$ times, irrespective of the number of sampling steps $D$. In the other steps, $\boldsymbol{x}_t$ remains in all $d$ dimensions. We highlight that we can cache $\boldsymbol{c}_\theta(\boldsymbol{x}_t)$ naturally without evaluating the time-independent $\boldsymbol{c}_\theta$ to reduce the NFE compared to SEDD (see Appendix E for the pseudo-code, ). As shown in Fig. 2, RADD with the caching strategy is more efficient than SEDD given any number of sampling steps, especially given large sampling steps. This is as expected because the NFE is limited within the generating sequence length.

Note that the NFEs with the caching strategy is a random variable. To quantify it, we calculate the expected NFEs (abbr. E-NFEs) required in an analytic form, conditioned on the sampling method, time steps, and noise schedule. Specifically, denote $l$ as the generating sequence length, which does not equal $d$ generally. Given the sampling time steps $\{t_0 = 0, \cdots, t_n = T\}$, let $N_k \in \{0, \cdots, l\}$ denote the number of changed dimensions of $\boldsymbol{x}$ in $[t_{k-1}, t_k)$. Since we perform function evaluation

in $[t_{k-1}, t_k)$ only when $\boldsymbol{x}$ changes (i.e. $N_k \neq 0$), the NFEs and E-NFEs can expressed as:

$$\text{NFEs}(n) = \sum_{k=1}^{n} \mathbb{I}(N_k \neq 0), \tag{3.3}$$

$$\text{E-NFEs}(n) = \sum_{k=1}^{n} \mathbb{E}[\mathbb{I}(N_k \neq 0)] = \sum_{k=1}^{n} P(N_k \neq 0). \tag{3.4}$$

For each dimension $i$, let $r_k$ represent the probability that $x^i$ changes within the interval $[t_{k-1}, t_k)$. As the probability is independent in different dimensions (proof in Appendix D.3), $N_k$ follows a binomial distribution with parameters $l$ and $r_k$. Therefore, Eq. (3.4) can be further simplified as:

$$\text{E-NFEs}(n) = \sum_{k=1}^{n} P(N_k \neq 0) = \sum_{k=1}^{n} (1 - (1 - r_k)^l), \tag{3.5}$$

which applies to all samplers. Further, $r_k$ can be analytically expressed w.r.t. the time steps and noise schedule for both Euler and Tweedie $\tau$-leaping samplers, as detailed in Appendix D.3. Taking Tweedie $\tau$-leaping method with log-linear noise schedule [29] for example, its E-NFEs is given by:

$$\text{E-NFEs}(n) = \sum_{k=1}^{n} (1 - (1 - \frac{1}{n})^l) = n(1 - (1 - \frac{1}{n})^l). \tag{3.6}$$

Appendix D.3 provides the proof. As shown in Fig. 1, we plot the curve of Eq. (3.6) in blue, which agrees with our experiments (the red stars).

## 3.3 Denoise cross-entropy for exact likelihood evaluation and training

As illustrated in Theorem 1, the concrete score can be understood as a rescaled conditional distribution on clean data. From this perspective, it is natural to wonder: **is it possible to evaluate and optimize the exact likelihood of the model instead of the ELBO?** Surprisingly, the answer is yes for both the original parameterization [29] and our new parameterization.

Let $q_\theta(\boldsymbol{x}_0)$ denote the model distribution at time zero defined by $\boldsymbol{s}_\theta$, or our $\boldsymbol{c}_\theta$, which approximates the true distribution $p_0(\boldsymbol{x}_0)$. Inspired by the cross-entropy loss in auto-regressive models, we define the denoising cross-entropy loss $\mathcal{L}_{\text{DCE}}^T(\boldsymbol{x}_0)$ as:

$$\mathcal{L}_{\text{DCE}}^T(\boldsymbol{x}_0) := \int_0^T \mathbb{E}_{\tilde{\boldsymbol{x}} \sim p_{t|0}(\cdot|\boldsymbol{x}_0)} \sum_{\boldsymbol{y} \neq \tilde{\boldsymbol{x}}} \boldsymbol{Q}_t(\tilde{\boldsymbol{x}}, \boldsymbol{y}) \left( -\frac{p_{t|0}(\boldsymbol{y} \mid \boldsymbol{x}_0)}{p_{t|0}(\tilde{\boldsymbol{x}} \mid \boldsymbol{x}_0)} \log \boldsymbol{s}_\theta(\tilde{\boldsymbol{x}}, t)_{\boldsymbol{y}} \right) dt, \tag{3.7}$$

$$= \int_0^T \mathbb{E}_{\tilde{\boldsymbol{x}} \sim p_{t|0}(\cdot|\boldsymbol{x}_0)} \sum_{\boldsymbol{y} \neq \tilde{\boldsymbol{x}}} \boldsymbol{Q}_t(\tilde{\boldsymbol{x}}, \boldsymbol{y}) \left( -\frac{p_{t|0}(\boldsymbol{y} \mid \boldsymbol{x}_0)}{p_{t|0}(\tilde{\boldsymbol{x}} \mid \boldsymbol{x}_0)} \log \left( \frac{e^{-\bar{\sigma}(t)}}{1 - e^{-\bar{\sigma}(t)}} \boldsymbol{c}_\theta(\tilde{x})_{\boldsymbol{y}} \right) \right) dt. \tag{3.8}$$

Compared with the DSE loss in Eq. (2.6), our DCE loss simply removed the terms $\boldsymbol{s}_\theta(\tilde{\boldsymbol{x}}, t)_{\boldsymbol{y}}$ and $K\left( \frac{p_{t|0}(\boldsymbol{y}|\boldsymbol{x}_0)}{p_{t|0}(\tilde{\boldsymbol{x}}|\boldsymbol{x}_0)} \right)$, however, it shows that DCE loss exactly equals the negative log-likelihood of $q_\theta(\boldsymbol{x}_0)$ with a sufficiently long process in absorbing discrete diffusion.

**Theorem 2.** *Suppose $\{X_t\}$ is a continuous time Markov chain with transition rate matrix $\boldsymbol{Q}_t = \sigma(t)\boldsymbol{Q}^{absorb}$. For a given data $\boldsymbol{x}_0$, if $\sigma(t)$ satisfies $\int_0^\infty \sigma(\tau)d\tau = \infty$, then the denoising cross-entropy loss defined in Eq. (3.8) with $T \to \infty$ exactly equals the negative log-likelihood of $\boldsymbol{x}_0$.*

$$\mathcal{L}_{DCE}^\infty(\boldsymbol{x}_0) = -\log q_\theta(\boldsymbol{x}_0). \tag{3.9}$$

The proof of Theorem 2 consists of three key steps, detailed in Appendix C.1, Appendix C.2 and Appendix C.3 respectively. In the first step, we apply a change of variable from $t$ to $\lambda(t) = 1 - e^{-\bar{\sigma}(t)}$, which is the probability of a token is masked from 0 to $t$ in the forward process. Further, inspired by the factorization form discovered in Theorem 1, the denoising cross-entropy loss for both parameterizations can then be rewritten as an integral of $\lambda$

$$\mathcal{L}_{\text{DCE}}^\infty(\boldsymbol{x}_0) = \int_0^1 \frac{1}{\lambda} \mathbb{E}_{\tilde{\boldsymbol{x}} \sim p_\lambda(\tilde{\boldsymbol{x}}|\boldsymbol{x}_0)} \left[ \sum_{\tilde{\boldsymbol{x}}^i = [\mathbf{M}]} -\log q_\theta(\boldsymbol{x}_0^i | \tilde{\boldsymbol{x}}^{\text{UM}}) \right] d\lambda, \tag{3.10}$$

where $p_\lambda(\tilde{x}|x_0)$ is the joint distribution induced by masking each dimension in $x_0$ independently with a probability $\lambda$.

In the second step, we demonstrate that the integral w.r.t. $\lambda$ in Eq. (3.10) can be integrated analytically, and the DSE loss can be rewritten as expectations over the number and positions of masks as follows:

$$\mathcal{L}_{\text{DCE}}^{\infty}(x_0) = d\mathbb{E}_{k \sim U(\{1,\cdots,d\})} \frac{1}{k} \mathbb{E}_{\tilde{x} \sim U(\tilde{\mathcal{X}}_k)} \left[ \sum_{\tilde{x}^i = [\mathbf{M}]} -\log q_\theta(x_0^i | \tilde{x}^{\text{UM}}) \right], \qquad (3.11)$$

where we denote $\tilde{\mathcal{X}}_k := \{\tilde{x} : \tilde{x} \in \tilde{\mathcal{X}} \text{ and } \tilde{x} \text{ has exact } k \text{ dimensions masked by } [\mathbf{M}]\}$ and $U(\cdot)$ as uniform distribution.

Finally, in the third step, we prove that Eq. (3.11) enumerates all orders to decompose the joint distribution auto-regressively and accumulates log densities of all conditional distributions in every order. Therefore, it is equivalent to the negative log-likelihood of $q_\theta$:

$$\mathcal{L}_{\text{DCE}}^{\infty}(x_0) = -\log q_\theta(x_0). \qquad (3.12)$$

Theorem 2 enables **exact likelihood computation** for both the original model $s_\theta$ and our $c_\theta$, providing a more accurate measure of model performance. Take $c_\theta$ for example, Eq. (3.8) can be rewritten as a form of expectation on $t$:

$$\mathcal{L}_{\text{DCE}}^{T}(x_0) = \frac{1}{T}\mathbb{E}_{t \sim U([0,T])}\mathbb{E}_{\tilde{x} \sim p_{t|0}(\cdot|x_0)} \sum_{y \neq \tilde{x}} Q_t(\tilde{x}, y)\left(-\frac{p_{t|0}(y \mid x_0)}{p_{t|0}(\tilde{x} \mid x_0)}\log\left(\frac{e^{-\bar{\sigma}(t)}}{1 - e^{-\bar{\sigma}(t)}}c_\theta(\tilde{x})_y\right)\right).$$
$$(3.13)$$

Naturally, we can take the Monte Carlo estimation of $\mathcal{L}_{\text{DCE}}^{T}(x_0)$ by sampling $t$ to approximate $-\log q_\theta(x_0)$ according to Eq. (3.13). In addition, it can be used as an efficient and valid training target for discrete diffusion models, as an alternative to the ELBO (i.e. DSE loss). For pseudo-code of training, see Appendix E.

# 4   Experiments

We present the experimental setups in Section 4.1. We then evaluate the performance of accelerated generation in Section 4.2 and zero-shot perplexity on various language datasets in Section 4.3.

## 4.1   Settings

**Model.**  We use RADD model $c_\theta$ reparameterzied as described in Section 3.1. Compared with SEDD small model, RADD model has 7M fewer parameters due to the removal of time-condition, which equates to an 8% decrease from the original 90M non-embedding parameters. We trained our RADD model $c_\theta$ using denoising score entropy and denoising cross entropy, abbreviated as RADD-DSE and RADD-DCE. For SEDD small model, we employed their pre-trained model.

**Data.**  In line with the methodology outlined by SEDD, we trained on the OpenWebText [33] dataset and tested on the LAMBADA, WikiText2, PTB, WikiText103, and One Billion Words datasets [34, 35, 36]. For data splits and data processing, we adopted the same settings and techniques as SEDD, which involves packing sentences to generate uniform-length blocks as model input.

**Training setup.**  We used the same training setup for RADD and SEDD. Specifically, we used a log-linear noise schedule where the expectation of the number of changed tokens at time $t$ is linear with $t$. For simplicity, we also used the same optimization configuration as SEDD, which can be suboptimal for our RADD model and DCE loss. For more details see Appendix F.

**Metric.**  Following previous work [29], we conduct experiments on unconditional generation and language modeling tasks. For generation, we use perplexity (PPL) on unconditional samples measured by an additional larger language model (i.e. GPT-2 large) to evaluate sample quality. To access inference efficiency, we computed the inference time on a single NVIDIA 4090 GPU with a batch size of 8 and averaged over 1024 samples. For language modeling tasks, we report the perplexity calculated on the dataset with different models.

Table 1: **Avarage inference time of a single sample with varying sampling steps.** The table compares the average inference time (in seconds) for the SEDD small model using both Euler and Tweedie $\tau$-leaping (abbreviated as T-$\tau$) sampling methods, and the RADD small model using the Euler method with a caching strategy.

| Methods | Metrics | 32 | 64 | 128 | 256 | 512 | 1024 | 2048 | 4096 |
|---|---|---|---|---|---|---|---|---|---|
| SEDD (euler) | Time(s) | 0.48 | 0.87 | 1.67 | 3.25 | 6.41 | 12.74 | 25.42 | 50.86 |
| | PPL | 155 | 105 | 81 | 66 | 53 | 43 | 35 | 28 |
| SEDD (T-$\tau$) | Time(s) | 0.38 | 0.68 | 1.28 | 2.47 | 4.85 | 9.61 | 19.14 | 38.20 |
| | PPL | 151 | 104 | 81 | 65 | 52 | 42 | 34 | 28 |
| RADD | Time(s) | **0.33** | **0.54** | **0.94** | **1.68** | **2.97** | **5.15** | **8.73** | **14.88** |
| | PPL | **135** | **94** | **72** | **58** | **46** | **37** | **30** | **26** |

## 4.2 Efficient sampling

We compare the sample quality measured by perplexity between SEDD and our RADD-DCE model, as shown in Fig. 2. For a fixed NFE, RADD-DCE with the Euler sampler outperforms SEDD with multiple samplers. It suggests that RADD with caching accelerates the sampling process and benefits sample quality at the same time. Besides, the acceleration by cache strategy is particularly significant with large sampling steps, as analyzed in Section 3.2.

We further compare the running time for the methods in Table 1. Across all sampling steps, RADD consistently requires the shortest sampling time and outperforms SEDD with different samplers. Quantitatively, RADD achieves a speed-up of $2.5 \sim 3.5$ times as shown in Table 1. These results agree with the analysis of the E-NFEs in Fig. 1, validate the effectiveness of RADD and caching strategy, and demonstrate the practical implications of our Theorem 1.

According to Eq. (3.11), we can also use RADD as an auto-regressive model to generate samples in different orders, leading to worse performance as a discrete diffusion, as detailed in Appendix F.4. We present more sampling details in Appendix F.3. and the generated samples in Appendix G.1.

## 4.3 Improved zero-shot perplexity on language modeling

Following SEDD, we present zero-shot perplexities on the LAMBADA, WikiText2, PTB, Wiki-Text103, and 1 Billion Words datasets [37] in Table 2 and compare the zero-shot perplexity of our model with other baseline models [20, 38, 29].

Firstly, we conduct an ablation study of the scaling trick in the middle of the Table 2. With an absorbing process, the perplexity of the scaled version of SEDD outperforms its unscaled version, which matches our theoretical discovery in Theorem 1.

Secondly, without any modification of the model, we estimate the exact likelihood of the baseline model SEDD [29] based on Theorem 2 in Table 2. We observe that perplexity is consistently better than the ELBO of the strongest discrete diffusion models, which validates our Theorem 2.

Lastly, we report the maximum likelihood training results of RADD in the last row in Table 2. We observed that RADD-DCE outperforms RADD-DSE, but their performances are slightly worse than SEDD. This discrepancy could be because we did not search the hyperparameters and directly applied identical optimization configures as SEDD, which may be suboptimal.

## 5 Related work

**Continouous-state diffusion models for text generation.** Several works have been proposed to apply continuous diffusion to text [19, 21, 22, 23]. Li et al. [19] use an embedding layer to map discrete tokens to a latent space and learn a continuous-state diffusion on it. Bit Diffusion [22] learns a continuous diffusion model to generate binary bits of discrete tokens. However, transforming between these continuous representations and discrete tokens by thresholding may lose information. Bayesian Flow Network [23] achieves competitive log-likelihood on character-level language modeling tasks

Table 2: **Zero-shot language modeling perplexity (↓) on five datasets.** [†] labels the results based on ELBO which is taken from [20, 38, 29] and [⋆] labels the results based on the exact likelihood implemented by us. In this table, SEDD-U / SEDD-S refer to the unscaled and scaled absorbing models respectively.

| Method | LAMBADA | WikiText2 | PTB | WikiText103 | 1BW |
|---|---|---|---|---|---|
| GPT-2 | **45.04** | 42.43 | 138.43 | 41.60 | **75.20** |
| D3PM[†] | ≤93.47 | ≤77.28 | ≤200.82 | ≤75.16 | ≤138.92 |
| PLAID[†] | ≤57.28 | ≤51.80 | ≤142.60 | ≤50.86 | ≤91.12 |
| SEDD-Uniform[†] | ≤65.40 | ≤50.27 | ≤140.12 | ≤49.60 | ≤101.37 |
| SEDD-U[†] | ≤52.21 | ≤44.75 | ≤130.49 | ≤43.14 | ≤80.70 |
| SEDD-S[†] | ≤50.92 | ≤41.84 | ≤114.24 | ≤40.62 | ≤79.29 |
| SEDD-S[⋆] (**Ours**) | 50.44 | **39.91** | **110.01** | **39.91** | 78.01 |
| RADD-DSE[⋆] (**Ours**) | 96.62 | 43.35 | 125.03 | 40.34 | 80.11 |
| RADD-DCE[⋆] (**Ours**) | 56.67 | 42.83 | 116.74 | 41.02 | 79.00 |

and is proven equivalent to continuous stochastic differential equations trained by denoising score matching [24]. Such models underperform auto-regressive models on standard text generation tasks.

**Discrete-state diffusion models for text generation.** Several discrete-state diffusion models have been proposed [11, 39, 20]. D3PM [20] proposed a diffusion framework based on any probability transition matrix and trained with a lower bound of log-likelihood. DiffusionBERT [25] utilizes a pre-trained BERT [40] as an initialization of diffusion. Furthermore, [26] generalizes the framework to continuous time by introducing a rate matrix. It is difficult to apply the score matching in such models because the gradient of the data distribution is undefined. Several works try to generalize the score matching on discrete data [29, 28, 26, 27]. Meng et al. [28] introduce the concrete score and the denoising concrete score matching loss. Furthermore, SEDD bridges the discrete state diffusion and the concrete score by introducing a denoising score entropy loss [29]. By incorporating an absorbing process, SEDD achieves competitive performance with the auto-regressive models, especially, GPT-2.

## 6 Conclusion

We introduce RADD, a dedicated discrete diffusion model that characterizes the time-independent conditional probabilities, built upon a new factorization form of the concrete score. RADD is much more efficient by reducing the NFEs with a cache strategy while retaining a better performance than strong baselines. Furthermore, we propose DCE loss and prove it is equivalent to the negative log-likelihood of absorbing diffusion. When applied to SEDD, DCE significantly advances the state-of-the-art discrete diffusion on 5 zero-shot language modeling benchmarks at the GPT-2 scale.

**Limitaition.** Our model has been trained and evaluated primarily on the GPT-2 scale. For broader applicability, it is essential to explore the effects of scaling on the performance [41], which is left as future work. The success of diffusion transformers on images [42, 43, 44] and videos [45] suggests that diffusion models can be scaled up by incorporating transformers.

Another limitation is that our model can only generate full-length outputs, unlike auto-regressive models that can produce variable-length outputs. This restricts the flexibility of our model in certain applications. We leave the investigation on this issue as future work.

**Social impact.** For the current theoretical and experimental scope of this paper, we have not found any direct social impacts. However, considering future developments, the paper potentially contributes to the next-generation large language models. In this context, this work could significantly reduce the inference cost of language models but may also lead to hallucinations, amplify biases and discrimination in the data, and pose risks of misuse. As with other generative models, addressing these issues requires further advancements in the field.

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

# A Proof of Proposition 1

Since the different dimensions of the forward diffusion process are independent of each other, we can first analyze the conditional distribution in one dimension. This can be derived directly from Eq. (2.3), but for a better understanding, here we provide a more intuitive proof in the case when $\boldsymbol{Q}_t = \boldsymbol{Q}^{\text{absorb}}$.

**Lemma 1.** *(Analytic conditional distribution for absorbing case) Suppose $\{X_t\}$ is a continuous time Markov chain with transition rate matrix $\boldsymbol{Q}_t = \sigma(t)\boldsymbol{Q}^{absorb}$, given the value $x$ at time zero , the conditional distribution $p_{t|0}(x_t|x)$ has the following analytic form:*

$$p_{t|0}(x_t|x) = \begin{cases} e^{-\bar{\sigma}(t)}, & x_t = x, \\ 1 - e^{-\bar{\sigma}(t)}, & x_t = [\boldsymbol{M}], \\ 0. & x_t \neq [\boldsymbol{M}] \text{ and } x_t \neq [\boldsymbol{M}]. \end{cases} \tag{A.1}$$

*Proof.* Given the initial value $x \in \mathcal{X} = \{1, \cdots, N\}$, we have

$$x_t = \begin{cases} x, & t < T_h, \\ [\mathbf{M}], & t \geq T_h, \end{cases}$$

where $T_h$ is the holding time before the system transitions to the next state.

Based on the properties of the $\boldsymbol{Q}^{\text{absorb}}$:

$$p_{t+\Delta t|t}(x|x) = 1 - (-\sigma(t)\boldsymbol{Q}^{\text{absorb}}(x,x))\Delta t + o(\Delta t). \tag{A.2}$$

Partitioning the interval $[0, t]$ into $\{s_k\}_{k=0}^n$ , make use of Memoryless Property of Continuous-Time Markov Chains:

$$p_{t|0}(x|x) = \prod_{k=1}^n p_{s_k|s_{k-1}}(x|x) \tag{A.3}$$

$$= \prod_{k=1}^n (1 - (-\sigma(t_k)\boldsymbol{Q}^{\text{absorb}}(x,x))(s_k - s_{k-1}) + o((s_k - s_{k-1}))) \tag{A.4}$$

$$= \exp(\sum_{k=1}^n \ln(1 - (-\sigma(t_k)\boldsymbol{Q}^{\text{absorb}}(x,x))(s_k - s_{k-1}) + o((s_k - s_{k-1}))) \tag{A.5}$$

$$= \exp(\sum_{k=1}^n -(-\sigma(t_k)\boldsymbol{Q}^{\text{absorb}}(x,x))(s_k - s_{k-1}) + o((s_k - s_{k-1}))). \tag{A.6}$$

Let $\max(s_k - s_{k-1}) \to 0$ , we have:

$$p_{t|0}(x|x) = \exp(-\int_0^t -\sigma(s)\boldsymbol{Q}^{\text{absorb}}(x,x)ds) = \exp(-(-\boldsymbol{Q}^{\text{absorb}}(x,x)\bar{\sigma}(t))). \tag{A.7}$$

As $\boldsymbol{Q}^{\text{absorb}}(x,x) = -1$, we have

$$p_{t|0}(x|x) = P(T_h > t) = e^{-\bar{\sigma}(t)}, \tag{A.8}$$

$$p_{t|0}([\mathbf{M}]|x) = P(T_h > t) = 1 - e^{-\bar{\sigma}(t)}, \tag{A.9}$$

$$p_{t|0}(k|x) = 0 \quad \text{if } k \neq [\mathbf{M}] \text{ and } k \neq x. \tag{A.10}$$

$\square$

**Proposition 1.** *(Analytic joint distribution for absorbing case)*

*Suppose $\{X_t\}$ is a continuous time Markov chain with transition rate matrix $\boldsymbol{Q}_t = \sigma(t)\boldsymbol{Q}^{absorb}$. For $\boldsymbol{x}_t = x_t^1 \cdots x_t^d$ with $N_1$ components as $[\boldsymbol{M}]$ and $N_2 = d - N_1$ components as specific value, $p_t(\boldsymbol{x}_t)$ can be expressed as Eq. (A.11):*

$$p_t(\boldsymbol{x}_t) = [1 - e^{-\bar{\sigma}(t)}]^{N_1} [e^{-\bar{\sigma}(t)}]^{N_2} p_0(\boldsymbol{x}_t^{UM}), \tag{A.11}$$

*where $\boldsymbol{x}_t^{UM} := \{\boldsymbol{x}^k | \boldsymbol{x}^k \neq [\boldsymbol{M}]\}$ represents unmasked part of $\boldsymbol{x}_t^{UM}$.*

Proposition 1 shows that the joint distribution $p_t(\boldsymbol{x}_t)$ can be expressed as the multiplication of two terms. One is an analytic term only depending on time, the other is a joint distribution of clean data $p_0(\boldsymbol{x}_t^{\text{UM}})$ with $N_2$ dimensions independent of time.

*Proof.* Without loss of generality, let's assume that the preceding $N_1$ terms of $\boldsymbol{x}$ are all $[\mathbf{M}]$, and the remaining $N_2$ terms are fixed at specific values. That is, $\boldsymbol{x}_t = [\mathbf{M}] \cdots [\mathbf{M}] x_t^{N_1+1} \cdots x_t^d$, and here $x^k$ is a fixed value in $\mathcal{X}$.

Use the law of total probability and Lemma 1, along with independent property:

$$p_t([\mathbf{M}] \cdots [\mathbf{M}] x_t^{N_1+1} \cdots x_t^d)$$
$$= \sum_{\boldsymbol{x}_0 \in \mathcal{X}^d} p_{t|0}([\mathbf{M}] \cdots [\mathbf{M}] x_t^{N_1+1} \cdots x_t^d | \boldsymbol{x}_0) p_0(\boldsymbol{x}_0)$$
$$= \sum_{x_0^1 \in \mathcal{X}, \cdots, x_0^d \in \mathcal{X}} p_{t|0}([\mathbf{M}] \cdots [\mathbf{M}] x_t^{N_1+1} \cdots x_t^d | x_0^1 \cdots x_0^d) p_0(x_0^1 \cdots x_0^d)$$
$$= \sum_{x_0^1 \in \mathcal{X}, \cdots, x_0^d \in \mathcal{X}} \prod_{k=1}^{N_1} p_{t|0}^k([\mathbf{M}]|x_0^k) \prod_{k=N_1+1}^{d} p_{t|0}^k(x_t^k|x_0^k) p_0(x_0^1 \cdots x_0^d)$$
$$= \sum_{x_0^1 \in \mathcal{X}, \cdots, x_0^{N_1} \in \mathcal{X}} \prod_{k=1}^{N_1} p_{t|0}^k([\mathbf{M}]|x_0^k)[e^{-\bar{\sigma}(t)}]^{N_2} p_0(x_0^1 \cdots x_0^{N_1} x_t^{N_1+1} \cdots x_t^d)$$
$$= \sum_{x_0^1 \in \mathcal{X}, \cdots, x_0^{N_1} \in \mathcal{X}} [1 - e^{-\bar{\sigma}(t)}]^{N_1} [e^{-\bar{\sigma}(t)}]^{N_2} p_0(x_0^1 \cdots x_0^{N_1} x_t^{N_1+1} \cdots x_t^d)$$
$$= [1 - e^{-\bar{\sigma}(t)}]^{N_1} [e^{-\bar{\sigma}(t)}]^{N_2} \sum_{x_0^1 \in \mathcal{X}, \cdots, x_0^{N_1} \in \mathcal{X}} p_0(x_0^1 \cdots x_0^{N_1} x_t^{N_1+1} \cdots x_t^d)$$
$$= [1 - e^{-\bar{\sigma}(t)}]^{N_1} [e^{-\bar{\sigma}(t)}]^{N_2} p_0(x_t^{N_1+1} \cdots x_t^d).$$

In the general case, we have:

$$p_t(\boldsymbol{x}_t) = [1 - e^{-\bar{\sigma}(t)}]^{N_1} [e^{-\bar{\sigma}(t)}]^{N_2} p_0(\boldsymbol{x}_t^{\text{UM}}),$$

which shows that the likelihood of noisy data $\boldsymbol{x}$ at time $t$ equals the likelihood of unmasked part of $\boldsymbol{x}$ at time 0 multiplied by a analytic time-dependent term. $\qquad\square$

# B  Proof of Theorem 1

**Theorem 1.** *(Analytic concrete score in absorbing case, proof in Appendix B) For $\boldsymbol{x}_t = x_t^1 \ldots x_t^i \ldots x_t^d$ and $\hat{\boldsymbol{x}}_t = x_t^1 \ldots \hat{x}_t^i \ldots x_t^d$, if $x_t^i = [\boldsymbol{M}]$ and $\hat{x}_t^i \neq [\boldsymbol{M}]$, the concrete score at time $t$ can be expressed as a* time-independent *conditional distribution at time zero multiplied by an analytic* time-dependent *term:*

$$\frac{p_t\left(x_t^1 \ldots \hat{x}_t^i \ldots x_t^d\right)}{p_t\left(x_t^1 \ldots x_t^i \ldots x_t^d\right)} = \frac{e^{-\bar{\sigma}(t)}}{1 - e^{-\bar{\sigma}(t)}} p_0(\hat{x}_t^i | \boldsymbol{x}_t^{\text{UM}}),$$

*where $\boldsymbol{x}_t^{\text{UM}}$ is the vector consists of all unmasked tokens of $\boldsymbol{x}_t$.*

*Proof.* According to Proposition 1, if $x_t^i = [\mathbf{M}]$ and $\hat{x}_t^i \neq [\mathbf{M}]$, $\hat{\boldsymbol{x}}_t^{\text{UM}} = (\boldsymbol{x}_t^{\text{UM}}, \hat{x}_t^i)$,

$$\frac{p_t(\hat{\boldsymbol{x}}_t)}{p_t(\boldsymbol{x}_t)} = \frac{[1 - e^{-\bar{\sigma}(t)}]^{N_1-1}[e^{-\bar{\sigma}(t)}]^{N_2+1} p_0(\hat{\boldsymbol{x}}_t^{\text{UM}})}{[1 - e^{-\bar{\sigma}(t)}]^{N_1}[e^{-\bar{\sigma}(t)}]^{N_2} p_0(\boldsymbol{x}_t^{\text{UM}})}$$
$$= \frac{[1 - e^{-\bar{\sigma}(t)}]^{N_1-1}[e^{-\bar{\sigma}(t)}]^{N_2+1} p_0(\boldsymbol{x}_t^{\text{UM}}, \hat{x}_t^i)}{[1 - e^{-\bar{\sigma}(t)}]^{N_1}[e^{-\bar{\sigma}(t)}]^{N_2} p_0(\boldsymbol{x}_t^{\text{UM}})}$$
$$= \frac{e^{-\bar{\sigma}(t)}}{1 - e^{-\bar{\sigma}(t)}} p_0(\hat{x}_t^i | \boldsymbol{x}_t^{\text{UM}}).$$

$\qquad\square$

# C Proof of Theorem 2

## C.1 Denoising cross-entropy loss by $\lambda$

According to definition of $\boldsymbol{Q}_t$ we can simplify Eq. (3.8) as:

$$
\mathcal{L}_{\mathrm{DCE}}^{\infty}(\boldsymbol{x}_0)
$$

$$
= \int_0^{\infty} \mathbb{E}_{\tilde{\boldsymbol{x}} \sim p_{t|0}(\tilde{\boldsymbol{x}}|\boldsymbol{x}_0)} \left[ \sum_{\tilde{x}^i=[\mathbf{M}], j \neq [\mathbf{M}]} \sigma(t) \left( -\frac{e^{-\bar{\sigma}(t)}}{1-e^{-\bar{\sigma}(t)}} I(x_0^i = j) \log \left( \frac{e^{-\bar{\sigma}(t)}}{1-e^{-\bar{\sigma}(t)}} \boldsymbol{c}_\theta(\tilde{\boldsymbol{x}})[i,j] \right) \right) \right] dt
$$

$$
= \int_0^{\infty} \mathbb{E}_{\tilde{\boldsymbol{x}} \sim p_{t|0}(\tilde{\boldsymbol{x}}|\boldsymbol{x}_0)} \left[ \sum_{\tilde{x}^i=[\mathbf{M}]} \sigma(t) \left( -\frac{e^{-\bar{\sigma}(t)}}{1-e^{-\bar{\sigma}(t)}} \log \left( \frac{e^{-\bar{\sigma}(t)}}{1-e^{-\bar{\sigma}(t)}} \boldsymbol{c}_\theta(\tilde{\boldsymbol{x}})[i,x_0^i] \right) \right) \right] dt
$$

$$
= \int_0^{\infty} \mathbb{E}_{\tilde{\boldsymbol{x}} \sim p_{t|0}(\tilde{\boldsymbol{x}}|\boldsymbol{x}_0)} \left[ \sum_{\tilde{x}^i=[\mathbf{M}]} \sigma(t) \left( -\frac{e^{-\bar{\sigma}(t)}}{1-e^{-\bar{\sigma}(t)}} \log \left( \frac{e^{-\bar{\sigma}(t)}}{1-e^{-\bar{\sigma}(t)}} q_\theta(x_0^i|\tilde{\boldsymbol{x}}^{\mathrm{UM}}) \right) \right) \right] dt.
$$

Define $\lambda(t) = 1 - e^{-\bar{\sigma}(t)}$, $d\lambda = \sigma(t)e^{-\bar{\sigma}(t)}dt$. As $\bar{\sigma}(t) = \int_0^t \sigma(\tau)d\tau$, we have $\lambda(0) = 0$,

$\lim_{t \to \infty} \lambda(t) = 1$. By a change of variables for the integration variable from $t$ to $\lambda$, we can

rewrite the above equation as:

$$
\int_0^1 \frac{1}{\lambda} \mathbb{E}_{\tilde{\boldsymbol{x}} \sim p_\lambda(\tilde{\boldsymbol{x}}|\boldsymbol{x}_0)} \left[ \sum_{\tilde{x}^i=[\mathbf{M}]} \left( -\log(\frac{1-\lambda}{\lambda} q_\theta(x_0^i|\tilde{\boldsymbol{x}}^{\mathrm{UM}})) \right) \right] d\lambda
$$

$$
= \int_0^1 \frac{1}{\lambda} \mathbb{E}_{\tilde{\boldsymbol{x}} \sim p_\lambda(\tilde{\boldsymbol{x}}|\boldsymbol{x}_0)} \sum_{\tilde{x}^i=[\mathbf{M}]} \left( -\log(q_\theta(x_0^i|\tilde{\boldsymbol{x}}^{\mathrm{UM}})) \right) d\lambda + \int_0^1 \frac{1}{\lambda} \mathbb{E}_{\tilde{\boldsymbol{x}} \sim p_\lambda(\tilde{\boldsymbol{x}}|\boldsymbol{x}_0)} \sum_{\tilde{x}^i=[\mathbf{M}]} \left( -\log(\frac{1-\lambda}{\lambda}) \right) d\lambda.
$$

By independence of forward process and Lemma 1, $p_{t|0}(\tilde{\boldsymbol{x}}|\boldsymbol{x}_0) = \prod_{i=1}^d p_{t|0}(\tilde{x}^i|x_0^i)$ where

$$
p_{t|0}(\tilde{x}^i|x_0^i) = \begin{cases} 1-e^{-\bar{\sigma}(t)} & \tilde{x}^i = [\mathbf{M}], \\ e^{-\bar{\sigma}(t)} & \tilde{x}^i = x_0^i, \\ 0 & \text{else.} \end{cases} \tag{C.1}
$$

Therefore, $p_\lambda(\tilde{\boldsymbol{x}}|\boldsymbol{x}_0) = \prod_{i=1}^d p_\lambda(\tilde{x}^i|x_0^i)$ where

$$
p_\lambda(\tilde{x}^i|x_0^i) = \begin{cases} \lambda & \tilde{x}^i = [\mathbf{M}], \\ 1-\lambda & \tilde{x}^i = x_0^i, \\ 0 & \text{else.} \end{cases} \tag{C.2}
$$

Consider the second term, we have:

$$
\int_0^1 \frac{1}{\lambda} \mathbb{E}_{\tilde{\boldsymbol{x}} \sim p_\lambda(\tilde{\boldsymbol{x}}|\boldsymbol{x}_0)} \left[ \sum_{\tilde{x}^i=[\mathbf{M}]} \left( -\log(\frac{1-\lambda}{\lambda}) \right) \right] d\lambda
$$

$$
= \int_0^1 \frac{1}{\lambda} \mathbb{E}_{\tilde{\boldsymbol{x}} \sim p_\lambda(\tilde{\boldsymbol{x}}|\boldsymbol{x}_0)} \left[ \sum_{i=1}^d \mathbb{I}(\tilde{x}^i = [\mathbf{M}]) \left( -\log(\frac{1-\lambda}{\lambda}) \right) \right] d\lambda
$$

$$
= \int_0^1 \frac{1}{\lambda} \left[ \sum_{i=1}^d p_\lambda(\tilde{x}^i = [\mathbf{M}]|x_0) \left( -\log(\frac{1-\lambda}{\lambda}) \right) \right] d\lambda
$$

$$
= d \int_0^1 -\log(\frac{1-\lambda}{\lambda}) d\lambda
$$

$$
= d \left( \lambda \log \lambda + (1-\lambda) \log(1-\lambda) \right) |_0^1.
$$

Note that:
$$
\lim_{\lambda \to 0} \lambda \log \lambda = \lim_{\lambda \to 1} (1-\lambda) \log(1-\lambda) = 0,
$$

therefore, $(\lambda \log \lambda + (1 - \lambda) \log(1 - \lambda))|_0^1 = 0$.

$$\mathcal{L}_{\text{DCE}}^{\infty}(\boldsymbol{x}_0) = \int_0^1 \frac{1}{\lambda} \mathbb{E}_{\tilde{\boldsymbol{x}} \sim p_\lambda(\tilde{\boldsymbol{x}}|\boldsymbol{x}_0)} \left[ \sum_{\tilde{x}^i = [\mathbf{M}]} \left( -\log(q_\theta(x_0^i|\tilde{\boldsymbol{x}}^{\text{UM}})) \right) \right] d\lambda. \tag{C.3}$$

## C.2  Denoising cross-entropy loss by $k$

By Eq. (C.3), we can express the loss in terms of $\lambda$. Given $x_0$, we denote $\tilde{\mathcal{X}} :=$ $\{x_0^1, [\mathbf{M}]\} \times \cdots \{x_0^d, [\mathbf{M}]\}$ as the sample space of $\tilde{x}$, and define $\tilde{\mathcal{X}}_k := \{\tilde{\boldsymbol{x}} : \tilde{\boldsymbol{x}} \in \tilde{\mathcal{X}} \wedge$ $\tilde{\boldsymbol{x}}$ has exact k dimensions with values$[\mathbf{M}]\}$. Obviously, $|\tilde{\mathcal{X}}| = 2^d$ and $|\tilde{\mathcal{X}}_k| = C_d^k$ . We have:

$$\int_0^1 \frac{1}{\lambda} \mathbb{E}_{\tilde{\boldsymbol{x}} \sim p_\lambda(\tilde{\boldsymbol{x}}|\boldsymbol{x}_0)} \left[ \sum_{\tilde{x}^i = [\mathbf{M}]} \left( -\log(q_\theta(x_0^i|\tilde{\boldsymbol{x}}^{\text{UM}})) \right) \right] d\lambda \tag{C.4}$$

$$= \int_0^1 \frac{1}{\lambda} \sum_{\tilde{x} \in \tilde{\mathcal{X}}} p_\lambda(\tilde{\boldsymbol{x}}|\boldsymbol{x}_0) \left[ \sum_{\tilde{x}^i = [\mathbf{M}]} \left( -\log(q_\theta(x_0^i|\tilde{\boldsymbol{x}}^{\text{UM}})) \right) \right] d\lambda \tag{C.5}$$

$$= \int_0^1 \frac{1}{\lambda} \sum_{k=0}^{d} \sum_{\tilde{\boldsymbol{x}} \in \tilde{\mathcal{X}}_k} \lambda^k (1 - \lambda)^{d-k} \left[ \sum_{\tilde{x}^i = [\mathbf{M}]} \left( -\log(q_\theta(x_0^i|\tilde{\boldsymbol{x}}^{\text{UM}})) \right) \right] d\lambda \tag{C.6}$$

$$= \int_0^1 \frac{1}{\lambda} \sum_{k=1}^{d} \sum_{\tilde{\boldsymbol{x}} \in \tilde{\mathcal{X}}_k} \lambda^k (1 - \lambda)^{d-k} \left[ \sum_{\tilde{x}^i = [\mathbf{M}]} \left( -\log(q_\theta(x_0^i|\tilde{\boldsymbol{x}}^{\text{UM}})) \right) \right] d\lambda \tag{C.7}$$

$$= \sum_{k=1}^{d} \int_0^1 \lambda^{k-1} (1 - \lambda)^{d-k} d\lambda \sum_{\tilde{\boldsymbol{x}} \in \tilde{\mathcal{X}}_k} \left[ \sum_{\tilde{x}^i = [\mathbf{M}]} \left( -\log(q_\theta(x_0^i|\tilde{\boldsymbol{x}}^{\text{UM}})) \right) \right] \tag{C.8}$$

$$= \sum_{k=1}^{d} \frac{(k-1)!(d-k)!}{d!} \sum_{\tilde{\boldsymbol{x}} \in \tilde{\mathcal{X}}_k} \left[ \sum_{\tilde{x}^i = [\mathbf{M}]} \left( -\log(q_\theta(x_0^i|\tilde{\boldsymbol{x}}^{\text{UM}})) \right) \right] \tag{C.9}$$

$$= \sum_{k=1}^{d} \frac{1}{k C_d^k} \sum_{\tilde{\boldsymbol{x}} \in \tilde{\mathcal{X}}_k} \left[ \sum_{\tilde{x}^i = [\mathbf{M}]} \left( -\log(q_\theta(x_0^i|\tilde{\boldsymbol{x}}^{\text{UM}})) \right) \right]. \tag{C.10}$$

This can be reformulated in the form of expectation:

$$\sum_{k=1}^{d} \frac{1}{k C_d^k} \sum_{\tilde{\boldsymbol{x}} \in \tilde{\mathcal{X}}_k} \left[ \sum_{\tilde{x}^i = [\mathbf{M}]} \left( -\log(q_\theta(x_0^i|\tilde{\boldsymbol{x}}^{\text{UM}})) \right) \right] \tag{C.11}$$

$$= \sum_{k=1}^{d} \frac{1}{k} \mathbb{E}_{\tilde{\boldsymbol{x}} \sim U(\tilde{\mathcal{X}}_k)} \left[ \sum_{\tilde{x}^i = [\mathbf{M}]} \left( -\log(q_\theta(x_0^i|\tilde{\boldsymbol{x}}^{\text{UM}})) \right) \right] \tag{C.12}$$

$$= d \mathbb{E}_{k \sim U(\{1,\cdots,d\})} \frac{1}{k} \mathbb{E}_{\tilde{\boldsymbol{x}} \sim U(\tilde{\mathcal{X}}_k)} \left[ \sum_{\tilde{x}^i = [\mathbf{M}]} \left( -\log(q_\theta(x_0^i|\tilde{\boldsymbol{x}}^{\text{UM}})) \right) \right]. \tag{C.13}$$

## C.3  Exact negative likelihood

Let $S_d$ represent the set of all permutations of the integers $1, \cdots, d$, and let $\pi \in S_d$ be one of these permutations. Then, we can express $\log q_\theta(\boldsymbol{x}_0)$ as follows:

$$\log q_\theta(\boldsymbol{x}_0) = \mathbb{E}_{\pi \sim U(S_d)} \log q_\theta(\boldsymbol{x}_0) \tag{C.14}$$

$$= \mathbb{E}_{\pi \sim U(S_d)} \sum_{l=1}^{d} \log q_\theta(x_0^{\pi(l)} | x_0^{\pi(<l)}) \tag{C.15}$$

$$= \sum_{l=1}^{d} \mathbb{E}_{\pi \sim U(S_d)} \log q_\theta(x_0^{\pi(l)} | x_0^{\pi(<l)}) \tag{C.16}$$

$$= \sum_{l=1}^{d} \frac{1}{d-l+1} \mathbb{E}_{\pi \sim U(S_d)} \sum_{r=l}^{d} \log q_\theta(x_0^{\pi(r)} | x_0^{\pi(<l)}) \tag{C.17}$$

$$= \sum_{k=1}^{d} \frac{1}{k} \mathbb{E}_{\pi \sim U(S_d)} \sum_{r=d-k+1}^{d} \log q_\theta(x_0^{\pi(r)} | x_0^{\pi(<d-k+1)}) \tag{C.18}$$

$$= d \mathbb{E}_{k \sim U(\{1,\cdots,d\})} \frac{1}{k} \mathbb{E}_{\pi \sim U(S_d)} \sum_{r=d-k+1}^{d} \log q_\theta(x_0^{\pi(r)} | x_0^{\pi(<d-k+1)}). \tag{C.19}$$

In this context, for a fixed $k$, the condition $x_0^{\pi(<d-k+1)}$ can be understood as the unmasked part of noisy data $\tilde{\boldsymbol{x}}^{\text{UM}}$. For $r = d - k + 1, \cdots, d$, $x_0^{\pi(r)}$ corresponds to the $k$ items of the masked part. Therefore, we have:

$$\mathbb{E}_{\pi \sim U(S_d)} \sum_{r=d-k+1}^{d} \log q_\theta(x_0^{\pi(r)} | x_0^{\pi(<d-k+1)}) = \mathbb{E}_{\tilde{\boldsymbol{x}} \sim U(\tilde{\mathcal{X}}_k)} \sum_{\tilde{x}^i = [\mathbf{M}]} \log q_\theta(x_0^i | \tilde{\boldsymbol{x}}^{\text{UM}}). \tag{C.20}$$

Thus, substituting back, we have:

$$-\log q_\theta(\boldsymbol{x}_0) = d \mathbb{E}_{k \sim U(\{1,\cdots,d\})} \frac{1}{k} \mathbb{E}_{\tilde{\boldsymbol{x}} \sim U(\tilde{\mathcal{X}}_k)} \sum_{\tilde{x}^i = [\mathbf{M}]} -\log q_\theta(x_0^i | \tilde{\boldsymbol{x}}^{\text{UM}}), \tag{C.21}$$

which is exactly Eq. (C.13).

This concludes the proof of the exact negative likelihood, showing the equivalence between the expected negative log-likelihood and the denoising cross-entropy formulation.

# D  Sampling methods of discrete diffusion

## D.1  Original form in discrete diffusion

**Euler discrete method**  According to the Eq. (2.7), take $t = s - \Delta s$ and use the Euler method, we can simulate the reverse process by iteratively taking small $\Delta t$ Euler steps at time s, calculate the reverse transition rate based on $\boldsymbol{s}_\theta(\boldsymbol{x}_s, s)$, and randomly sampling $\boldsymbol{x}_{s-\Delta s}$.

Left term:
$$\frac{d}{dt} \boldsymbol{P}_{s \to t}\big|_{t=s-\Delta s} \approx \frac{\boldsymbol{P}_{s \to s-\Delta s} - \boldsymbol{P}_{s \to s}}{\Delta s}. \tag{D.1}$$

Right term:
$$\boldsymbol{P}_{s \to t} \tilde{\boldsymbol{Q}}_t\big|_{t=s-\Delta s} \approx \boldsymbol{P}_{s \to s} \tilde{\boldsymbol{Q}}_s. \tag{D.2}$$

As $\boldsymbol{P}_{s \to s} = \boldsymbol{I}$, we have

$$\boldsymbol{P}_{s \to s-\Delta s} \approx \boldsymbol{I} + \tilde{\boldsymbol{Q}}_s \Delta s. \tag{D.3}$$

Rewrite in $t$ and consider a specific input $x_t$, $x_{t-\Delta t}$ is sampled from the following transition probabilities:

$$p_{t-\Delta t|t}(x_{t-\Delta t}|x_t) \approx \delta_{x_t x_{t-\Delta t}} + \tilde{\boldsymbol{Q}}_t(x_t, x_{t-\Delta t})\Delta t + O(\Delta t) \tag{D.4}$$

$$\approx \delta_{x_t x_{t-\Delta t}} + \tilde{\boldsymbol{Q}}_t(x_t, x_{t-\Delta t})\Delta t, \tag{D.5}$$

where

$$\tilde{\boldsymbol{Q}}_t(x_t, x_{t-\Delta t}) \approx \begin{cases} \boldsymbol{Q}_t(x_{t-\Delta t}, x_t)\boldsymbol{s}_\theta(x_t, t)_{x_{t-\Delta t}} & x_t \neq x_{t-\Delta t}, \\ -\sum_{k \neq x_t} \tilde{\boldsymbol{Q}}_t(x_t, k) & x_t = x_{t-\Delta t}. \end{cases} \tag{D.6}$$

**Tweedie $\tau$ -leaping** If we know the analytic form of $\boldsymbol{P}_{s \to t}$, it is possible to get the closed form of reverse probability $\boldsymbol{P}_{t \to s}$ for any $s < t$. According to the conditional decomposition of total probability, we have:

$$\mathrm{diag}\left(P_t^T\right)\boldsymbol{P}_{t \to s} = \left(\mathrm{diag}\left(P_s^T\right)\boldsymbol{P}_{s \to t}\right)^T. \tag{D.7}$$

As $P_s^T \boldsymbol{P}_{s \to t} = \boldsymbol{P}_t^T$, the following equation holds:

$$\boldsymbol{P}_{t \to s} = \mathrm{diag}\left(P_t^T\right)^{-1}\boldsymbol{P}_{s \to t}^T \mathrm{diag}\left(P_s^T\right) = \mathrm{diag}\left(P_t^T\right)^{-1}\boldsymbol{P}_{s \to t}^T \mathrm{diag}\left(P_t^T \boldsymbol{P}_{s \to t}^{-1}\right). \tag{D.8}$$

Given $x_t$, to get $p_{s|t}(x_s|x_t)$, we only need to calculate row $x_t$ of $P_{t \to s}$ :

$$\boldsymbol{P}_{t \to s}(x_t, \cdot) = \frac{1}{p_t(x_t)}\boldsymbol{P}_{s \to t}^T(x_t, \cdot) \odot (P_t^T \boldsymbol{P}_{s \to t}^{-1}) \tag{D.9}$$

$$= \boldsymbol{P}_{s \to t}^T(x_t, \cdot) \odot \left(\frac{P_t^T}{p_t(x_t)}\boldsymbol{P}_{s \to t}^{-1}\right) \approx \boldsymbol{P}_{s \to t}^T(x_t, \cdot) \odot (\boldsymbol{s}_\theta(x_t, t)^T \boldsymbol{P}_{s \to t}^{-1}). \tag{D.10}$$

## D.2 Simplified form in reparameterized absorbing discrete diffusion

**Euler discrete method** For $x_t = [\mathbf{M}]$, given the value of $\hat{x}_t$, use the $\boldsymbol{Q}_t(\hat{x}_t, x_t) = \sigma(t)\boldsymbol{Q}^{\mathrm{absorb}}(\hat{x}_t, x_t)$ and $\boldsymbol{s}_\theta(x_t, t)_{\hat{x}_t} = \frac{e^{-\bar{\sigma}(t)}}{1 - e^{-\bar{\sigma}(t)}}\boldsymbol{c}_\theta(x_t)_{\hat{x}_t}$. Eq. (D.5) can be simplified as:

$$p_{t-\Delta t|t}(\hat{x}_t|[\mathbf{M}]) = \begin{cases} \sigma(t)\frac{e^{-\bar{\sigma}(t)}}{1 - e^{-\bar{\sigma}(t)}}\Delta t \boldsymbol{c}_\theta(x_t)_{\hat{x}_t} & \text{if } \hat{x}_t \neq [\mathbf{M}], \\ 1 - \sigma(t)\frac{e^{-\bar{\sigma}(t)}}{1 - e^{-\bar{\sigma}(t)}}\Delta t & \text{if } \hat{x}_t = [\mathbf{M}]. \end{cases} \tag{D.11}$$

For multi-dimension cases, similar results can be obtained:

$$p_{t-\Delta t|t}(x_{t-\Delta t}^i|\boldsymbol{x}_t) = \begin{cases} \sigma(t)\frac{e^{-\bar{\sigma}(t)}}{1 - e^{-\bar{\sigma}(t)}}\Delta t \boldsymbol{c}_\theta(\boldsymbol{x}_t)[i, x_{t-\Delta t}^i] & \text{if } x_{t-\Delta t}^i \neq [\mathbf{M}], \\ 1 - \sigma(t)\frac{e^{-\bar{\sigma}(t)}}{1 - e^{-\bar{\sigma}(t)}}\Delta t & \text{if } x_{t-\Delta t}^i = [\mathbf{M}]. \end{cases} \tag{D.12}$$

for all $x_t^i = [\mathbf{M}]$.

**Tweedie $\tau$-leaping** Suppose $\boldsymbol{x}_t = x_t^1 \cdots x_t^d$ has $N_1$ components as $[\mathbf{M}]$ and $N_2 = d - N_1$ components as specific values. Without loss of generality, let's assume that the preceding $N_1$ terms of $\boldsymbol{x}_t$ are all $[\mathbf{M}]$, and the remaining $N_2$ terms are fixed at specific values. For $1 \leq i \leq d$, given the value of $x_{t-\Delta t}^i \neq [\mathbf{M}]$, :

$$p_{t-\Delta t|t}(x_{t-\Delta t}^i|\boldsymbol{x}_t) = \frac{p_{t,t-\Delta t}(\boldsymbol{x}_t, x_{t-\Delta t}^i)}{p_t(\boldsymbol{x}_t)}. \tag{D.13}$$

By Proposition 1:

$$p_t(\boldsymbol{x}_t) = [1 - e^{-\bar{\sigma}(t)}]^{N_1}[e^{-\bar{\sigma}(t)}]^{N_2}p_0(x_t^{N_1+1} \cdots x_t^d). \tag{D.14}$$

531 Similar to the proof in Proposition 1:

$$p_{t,t-\Delta t}(\boldsymbol{x}_t, x^i_{t-\Delta t})$$

$$= \sum_{x_0 \in \mathcal{X}} p_{(t,t-\Delta t)|0} p(\boldsymbol{x}_t, x^i_{t-\Delta t}|\boldsymbol{x}_0) p_0(\boldsymbol{x}_0)$$

$$= \sum_{x_0^1 \in \mathcal{X}, \cdots, x_0^d \in \mathcal{X}} p_{(t,t-\Delta t)|0}([\mathbf{M}] \cdots [\mathbf{M}] x_t^{N_1+1} \cdots x_t^d, x^i_{t-\Delta t}|x_0^1 \cdots x_0^d) p_0(x_0^1 \cdots x_0^d)$$

$$= \sum_{x_0^1 \in \mathcal{X}, \cdots, x_0^d \in \mathcal{X}} p_{(t,t-\Delta t)|0}([\mathbf{M}], x^i_{t-\Delta t}|x_0^i) \prod_{k=1, k \neq i}^{N_1} p_{t|0}([\mathbf{M}]|x_0^k) \prod_{k=N_1+1}^{d} p_{t|0}(x^k|x_0^k) p_0(x_0^1 \cdots x_0^d)$$

$$= \sum_{x_0^k \in \mathcal{X}, k \in \{1, \cdots, N_1\}/\{i\}} p_{(t,t-\Delta t)|0}([\mathbf{M}], x^i_{t-\Delta t}|x^i_{t-\Delta t}) \prod_{k=1, k \neq i}^{N_1} p_{t|0}([\mathbf{M}]|x_0^k) [e^{-\bar{\sigma}(t)}]^{N_2}$$

$$p_0(x_0^1 \cdots x_0^{i-1} x^i_{t-\Delta t} x_0^{i+1} \cdots x_0^{N_1} x_t^{N_1+1} \cdots x_t^{N_1+1})$$

$$= \sum_{x_0^k \in \mathcal{X}, k \in \{1, \cdots, N_1\}/\{i\}} (e^{-\bar{\sigma}(t-\Delta t)} - e^{-\bar{\sigma}(t)})(1 - e^{-\bar{\sigma}(t)})^{N_1-1} [e^{-\bar{\sigma}(t)}]^{N_2}$$

$$p_0(x_0^1 \cdots x_0^{i-1} x^i_{t-\Delta t} x_0^{i+1} \cdots x_0^{N_1} x_t^{N_1+1} \cdots x_t^d)$$

$$= (e^{-\bar{\sigma}(t-\Delta t)} - e^{-\bar{\sigma}(t)})(1 - e^{-\bar{\sigma}(t)})^{N_1-1} [e^{-\bar{\sigma}(t)}]^{N_2} p_0(x^i_{t-\Delta t}, x_t^{N_1+1} \cdots x_t^d).$$

532 Note we used the fact that:

$$p_{(t,t-\Delta t)|0}([\mathbf{M}], x^i_{t-\Delta t}|x^i_{t-\Delta t}) = p_{t|t-\Delta t}([\mathbf{M}]|x^i_{t-\Delta t}) p_{t-\Delta t|0}(x^i_{t-\Delta t}|x^i_{t-\Delta t})$$

$$= (1 - e^{-(\bar{\sigma}(t) - \bar{\sigma}(t-\Delta t))}) e^{-\bar{\sigma}(t-\Delta t)}$$

$$= e^{-\bar{\sigma}(t-\Delta t)} - e^{-\bar{\sigma}(t)},$$

533

$$p_{t|0}([\mathbf{M}]|x_0^k) = 1 - e^{-\bar{\sigma}(t)},$$

534 by dividing the two expressions, we have:

$$p_{t-\Delta t|t}(x^i_{t-\Delta t}|\boldsymbol{x}_t) = \frac{e^{-\bar{\sigma}(t-\Delta t)} - e^{-\bar{\sigma}(t)}}{1 - e^{\bar{\sigma}(t)}} p_0(x^i_{t-\Delta t}|x_t^{N_1+1} \cdots x_t^d) \tag{D.15}$$

$$\approx \frac{e^{-\bar{\sigma}(t-\Delta t)} - e^{-\bar{\sigma}(t)}}{1 - e^{\bar{\sigma}(t)}} \boldsymbol{c}_\theta(\boldsymbol{x}_t)[i, x^i_{t-\Delta t}]. \tag{D.16}$$

535 In general, for $x_t^i = [\mathbf{M}]$, we have:

$$p_{t-\Delta t|t}(x^i_{t-\Delta t}|\boldsymbol{x}_t) \begin{cases} \approx \frac{e^{-\bar{\sigma}(t-\Delta t)} - e^{-\bar{\sigma}(t)}}{1 - e^{\bar{\sigma}(t)}} \boldsymbol{c}_\theta(\boldsymbol{x}_t)[i, x^i_{t-\Delta t}], & x^i_{t-\Delta t} \neq [\mathbf{M}], \\ = \frac{1 - e^{-\bar{\sigma}(t-\Delta t)}}{1 - e^{-\bar{\sigma}(t)}}, & x^i_{t-\Delta t} = [\mathbf{M}]. \end{cases} \tag{D.17}$$

### D.3 Discuss on the expectation of NFE

537 As discussed in Section Appendix D.2, for both the Euler method and Tweedie $\tau$-leaping, the
538 probability $p^i_{t-\Delta t|t}([\mathbf{M}]|\boldsymbol{x}_t)$ is only a factor of time which is independent of the other dimensions
539 of $\boldsymbol{x}_t$ once given $x_t^i = [\mathbf{M}]$. By the Law of Total Probability, it is easy to find that $p^i_{t-\Delta t|t}([\mathbf{M}]|[\mathbf{M}])$
540 is also only a factor of time. Thus, given a specific sampling method and a set of time steps
541 $\{t_0 = 0, \cdots, t_n = T\}$, the NFE can be treated as a random variable with a calculable expected value.

542 Let $N_k$ denote the number of dimensions of $\boldsymbol{x}$ which changed in $[t_{k-1}, t_k)$, so we have:

$$\text{NFEs}(n) = \sum_{k=1}^{n} \mathbb{I}(N_k \neq 0), \tag{D.18}$$

543

$$\text{E-NFEs}(n) = \sum_{k=1}^{n} \mathbb{E}[\mathbb{I}(N_k \neq 0)] = \sum_{k=1}^{n} P(N_k \neq 0). \tag{D.19}$$

For each dimension $i$, let $r_k$ represent the probability that $x^i$ changes within the interval $[t_{k-1}, t_k]$. Consequently, $N_k$ follows a binomial distribution with parameters $l$ and $r_k$, denoted as $N_k \sim Binomial(l, r_k)$.

$$\text{E-NFEs}(n) = \sum_{k=1}^{n} P(N_k \neq 0) = \sum_{k=1}^{n}(1 - (1 - r_k)^l). \tag{D.20}$$

By definition of $r_k$ and property of absorbing diffusion:

$$r_k = P(X_{t_{k-1}}^i \neq [\mathbf{M}], X_{t_k}^i = [\mathbf{M}] | X_{t_n}^i = [\mathbf{M}]) \tag{D.21}$$

$$= P(X_{t_{k-1}}^i \neq [\mathbf{M}] | X_{t_k}^i = [\mathbf{M}]) \prod_{l=k+1}^{n} P(X_{t_{l-1}}^i = [\mathbf{M}] | X_{t_l}^i = [\mathbf{M}]) \tag{D.22}$$

$$= (1 - P(X_{t_{k-1}}^i = [\mathbf{M}] | X_{t_k}^i = [\mathbf{M}])) \prod_{l=k+1}^{n} P(X_{t_{l-1}}^i = [\mathbf{M}] | X_{t_l}^i = [\mathbf{M}]). \tag{D.23}$$

Eq. (D.23) can be determined given the sampling method and noise schedule.

For the Euler method, based on Equation Eq. (D.12), we can derive that:

$$P(X_{t_{l-1}}^i = [\mathbf{M}] | X_{t_l}^i = [\mathbf{M}]) = 1 - \sigma(t_l) \frac{e^{-\bar{\sigma}(t_l)}}{1 - e^{-\bar{\sigma}(t_l)}}(t_l - t_{l-1}). \tag{D.24}$$

Therefore, we can express $r_k$ as:

$$r_k = (\sigma(t_k) \frac{e^{-\bar{\sigma}(t_k)}}{1 - e^{-\bar{\sigma}(t_k)}}(t_k - t_{k-1})) \prod_{l=k+1}^{n} (1 - \sigma(t_l) \frac{e^{-\bar{\sigma}(t_l)}}{1 - e^{-\bar{\sigma}(t_l)}}(t_l - t_{l-1})). \tag{D.25}$$

For Tweedie $\tau$-leaping, By Eq. (D.17), similarly we have:

$$P(X_{t_{l-1}}^i = [\mathbf{M}] | X_{t_l}^i = [\mathbf{M}]) = \frac{1 - e^{-\bar{\sigma}(t_{l-1})}}{1 - e^{-\bar{\sigma}(t_l)}}, \tag{D.26}$$

$$r_k = (\frac{e^{-\bar{\sigma}(t_{k-1})} - e^{-\bar{\sigma}(t_k)}}{1 - e^{-\bar{\sigma}(t_k)}}) \prod_{l=k+1}^{n} (1 - \frac{1 - e^{-\bar{\sigma}(t_{l-1})}}{1 - e^{-\bar{\sigma}(t_l)}}) = \frac{e^{-\bar{\sigma}(t_{k-1})} - e^{-\bar{\sigma}(t_k)}}{1 - e^{-\bar{\sigma}(t_n)}}. \tag{D.27}$$

Specifically, if we adopt a log-linear noise schedule, which implies $\bar{\sigma}(t) = -\log(1 - (1 - \epsilon)t)$ and $t_k = \frac{k}{n}$, Equation Eq. (D.27) can be simplified to $\frac{1}{n}$. Substituting this result into Equation Eq. (D.20), we obtain:

$$\text{E-NFEs}(n) = \sum_{k=1}^{n}(1 - (1 - \frac{1}{n})^l) = n(1 - (1 - \frac{1}{n})^l). \tag{D.28}$$

# E    Algorithms for training and inference

# F    Experimental details

## F.1    Model details

We implemented our RADD model based on SEDD architecture, which is an encoder-only transformer model [46, 47] incorporating time conditioning [48] and using rotary positional encoding [49]. The only difference is that we removed all parts related to time conditioning (i.e. TimeEmbedding, adaLN-zero block [48]) and added a softmax operation at the end of the neural network to ensure the output was a valid conditional distribution. Compared with SEDD small model, this modification led to a reduction of 7M parameters, equating to an 8% decrease from the original 90M non-embedding parameters.

---
**Algorithm 1** Unconditional Sampling
---
**Require:** Network $c_\theta$, noise schedule $\sigma$ (total noise $\bar{\sigma}$), time range $[0, T]$, step size $\Delta t$
1: $t \leftarrow T$, $\boldsymbol{x}_T \leftarrow \underbrace{[\mathbf{M}] \dots [\mathbf{M}]}_{d \times [\mathbf{M}]}$, $\boldsymbol{c}_{cache} \leftarrow \boldsymbol{c}_\theta(\boldsymbol{x}_t)$
2: **while** $t > 0$ **do**
3:     **if** Use Euler **then**
4:         Construct transition densities $p(x_{t-\Delta t}^i | \boldsymbol{x}_t)$ by Eq. (D.12) use $\boldsymbol{c}_{cache}$
5:     **end if**
6:     **if** Use Tweedie $\tau$ -leaping **then**
7:         Construct transition densities $p(x_{t-\Delta t}^i | \boldsymbol{x}_t)$ by Eq. (D.17) use $\boldsymbol{c}_{cache}$
8:     **end if**
9:     $x_{t-\Delta t}^i \sim \text{Cat}(p(x_{t-\Delta t}^i | x_t))$ for all $x_t^i = [\mathbf{M}]$, $x_{t-\Delta t}^i \leftarrow x_t^i$ for all $x_t^i \neq [\mathbf{M}]$
10:     **if** $\boldsymbol{x}_{t-\Delta t} \neq \boldsymbol{x}_t$ **then**
11:         $\boldsymbol{c}_{cache} \leftarrow \boldsymbol{c}_\theta(\boldsymbol{x}_t)$
12:     **end if**
13:     $t \leftarrow t - \Delta t$,
14: **end while**
---

---
**Algorithm 2** Conditional Sampling
---
**Require:** Network $c_\theta$, noise schedule $\sigma$ (total noise $\bar{\sigma}$), time range $[0, T]$, step size $\Delta t$, Prompt spaces $\Omega$ and tokens $\mathcal{T}$ .
1: $t \leftarrow T$, construct $\boldsymbol{x}_T$ with $\boldsymbol{x}_T^\Omega = \mathcal{T}$ and $\boldsymbol{x}_T^{\bar{\Omega}} = [\mathbf{M}]$, $\boldsymbol{c}_{cache} \leftarrow \boldsymbol{c}_\theta(\boldsymbol{x}_t)$
2: **while** $t > 0$ **do**
3:     **if** Use Euler **then**
4:         Construct transition densities $p(x_{t-\Delta t}^i | \boldsymbol{x}_t)$ by Eq. (D.12) use $\boldsymbol{c}_{cache}$
5:     **end if**
6:     **if** Use Tweedie $\tau$ -leaping **then**
7:         Construct transition densities $p(x_{t-\Delta t}^i | \boldsymbol{x}_t)$ by Eq. (D.17) use $\boldsymbol{c}_{cache}$
8:     **end if**
9:     $x_{t-\Delta t}^i \sim \text{Cat}(p(x_{t-\Delta t}^i | x_t))$ for all $x_t^i = [\mathbf{M}]$, $x_{t-\Delta t}^i \leftarrow x_t^i$ for all $x_t^i \neq [\mathbf{M}]$
10:     **if** $\boldsymbol{x}_{t-\Delta t} \neq \boldsymbol{x}_t$ **then**
11:         $\boldsymbol{c}_{cache} \leftarrow \boldsymbol{c}_\theta(\boldsymbol{x}_t)$
12:     **end if**
13:     $t \leftarrow t - \Delta t$,
14: **end while**
---

---
**Algorithm 3** Training
---
**Require:** Network $c_\theta$, noise schedule $\sigma$ (total noise $\bar{\sigma}$), time range $[0, T]$, data distribution $p_{\text{data}}$
1: **repeat**
2:     $\boldsymbol{x}_0 \sim p_{\text{data}}$, $t \sim U([0, T])$.
3:     construct $\boldsymbol{x}_t$ by $Z^i \sim Bernoulli(e^{-\bar{\sigma}(t)})$, $x_t^i = \mathbb{I}(Z^i = 1)x_0^i + \mathbb{I}(Z^i = 0)[\mathbf{M}]$
4:     Calculate $L_\theta(\boldsymbol{x}_t, \boldsymbol{x}_0) = \sum_{x_t^i = [\mathbf{M}]} -\sigma(t) \frac{e^{-\bar{\sigma}(t)}}{1 - e^{-\bar{\sigma}(t)}} \log\left( \frac{e^{-\bar{\sigma}(t)}}{1 - e^{-\bar{\sigma}(t)}} \boldsymbol{c}_\theta(\boldsymbol{x}_t)[i, x_0^i] \right)$
5:     Take gradient descent on $\nabla_\theta L(x_t, x_0)$
6: **until** converged
---

Table 3: **Quality of unconditionally generated text evaluated by perplexity ($\downarrow$).** For a fixed model, the best perplexity is **bolded**.

| Method | RADD-DSE | RADD-DCE |
|---|---|---|
| Forward | 116.94 | 113.92 |
| Backward | 135.39 | 125.59 |
| Random | **114.94** | **101.23** |

## F.2 Training details

Following the settings in [29], we trained our model with the following configuration:

- Batch Size:512
- Learning Rate: $3 \times 10^{-4}$
- Exponential Moving Average (EMA):0.9999
- Gradient Clipping: Gradient norm clipped to 1
- Warmup Schedule: Applied for the first 2500 iterations

We utilized 16 V100 32G GPUs or 16 A100 40G GPUs for training. For the A100 40G GPUs, we leveraged flash attention to accelerate the training process. For the V100 32G GPUs, which do not support flash attention or bfloat16, we employed float16 precision and used the Memory-Efficient Attention mechanism available in torch.nn.functional.scaled_dot_product_attention. Additionally, we used gradient checkpointing technique to save memory.

## F.3 Unconditional generation details

We used Tweedie $\tau$-leaping method, which has optimal results with fixed NFE. For SEDD small, we directly used their result. For RADD small, we generated 1000 samples to get the average value following [29].

## F.4 Further evaluation of generative perplexity

As stated in Theorem 1, $c_\theta$ can be interpreted as a conditional distribution over clean data. A natural idea is to use it directly to generate samples, which is similar to auto-regressive models. However, there are $d!$ kinds of decomposition from joint distribution to conditional distribution, in which we only tested three representative cases:

- forward: $p(x^1 \cdots x^d) = \prod_{k=1}^d p(x^k | x^{(<k)})$
- backward: $p(x^1 \cdots x^d) = \prod_{k=1}^d p(x^k | x^{(>k)})$
- random: $\pi \sim U(S_d), p(x^1 \cdots x^d) = \prod_{k=1}^d p(x^{\pi(k)} | x^{\pi(<k)})$

Results are shown in Table 3. The perplexity is calculated on average of 1024 samples. For the random case, we calculate the average perplexity between different randomly generated $\pi$. Generally, we find that the perplexity by directly sampling from the conditional distribution is higher than that achieved by Tweedie $\tau$-leaping. Among the different decomposition orders, the random order demonstrated the best performance.

# G Additional experimental results

## G.1 Additional samples

In this section, we present the unconditionally and conditionally generated text of RADD-DSE in Fig.3 and Fig.4, respectively. Similarly, the results of RADD-DCE are shown in Fig.**??** and Fig.**??**, respectively.

and human face. "And pretty damn conventional mating, so didn't come to the table like that with me. (As a character), it would be a pretty solid case to have," Andra says. "I saw the way he did it, and I think played a little bit with some of his fans. He wanted me to get cute a little bit too."

Advertisement

As in the parking lot, Andra was growing frustrated with the way the cars recently fit in nicely.

It also makes me feel like some things haven't changed until around this period of time, in the future. It's definitely the future here now — and I'm always dubious about thinking quite long before Toyota ever introduced a new car. "I think something that perfectly well fits all the guys," Andra says. "I therefore could fit more well."

– Follow Matt Dyckton on Twitter @Mittington.<|endoftext|>Spanish star Christina Rene got away after a teenage girl told to leave India for an unfamiliar place.

The Russian woman chose Chelsea to stay home, saying the alternative was to get rid of a condition and die after receiving a medical diagnosis and go back to learning as a nurse.

Chelsea was sentenced to the first arrest for a serious mental health conviction in October, and was transferred to a man who had taken his place, a 16-year-old man. The bailee's Appeal Court had appealed to a court to hear the case she learned from the teenagers.

The Russian Internal Revenue Service found the woman charged with administering an emergency ward, and although the girl was still waiting for a doctor there, she instead went to visit another clinic for the treatment of female swi-virus virus.

The court had not ruled out an in-life medical professional. Chelsea never applied to be a pregnant mother; her formal application assumed she was pregnant, dating from the summer of 2014.

Out of sound doubts, when her co-boyfriend Chelsea received a proposal to stay abroad for summer work abroad. Chelsea then applied to win to work and have a place in Russia.

The grant of nearly $5000 Chelsea invited Chelsea out to go see a Moscow clinic. It took two weeks to find a doctor. These sors' of Moscow's federal courts confiscated the grant.

The 17-year-old did not need to go to the hospital where she says she has received everything she has had in China, of course. Chelsea's lawyers Andre James Irani gave the court the doctors he could plead without permission of the young teenager on a regular basis.

The Argentine was put out on bail, prompting a sex psychologist to meet her when she signed her papers, but no family member was present.

"First thing I wasn't going to ask three days, but I'm thinking about this months already. "I'm glad to see that they are waiting for her with their services. She's already built a good life. She's interested in her studies. But feel like she is? I want to be the only person who has ever been my friend," she said.

"I took a lot of act, but she is more than never."

To this day Christina Rene McCourner told the court the situation is between Denmark and Stockholm syndrome. She says Chelsea has refused to come to term.

"She says that she is talented at medical school. But she's telling a different story, anyway. There's also a case when you just can't get the CNN-type cowardice yourself going to the doctor," she said.

A team of Barcelona has been conducting medical inquiries into both doctors who treat the sick and those not who use medical services. "How has she been following her visit to Russia United States and since arriving unable to do so?"

"Each often when she said "All the bills are what may I doctor," I've usually never used my medicine again," Chelsea attorneys say.

Life in the wrong world

While far-so in the past two weeks though, nearly speaking only three pregnant women in Mexico and Europe have requested that Chelsea stop using medical services at them all. Though many of its applicants have receiving medical assistance in Qatar and other parts of the world, they believe it means they should go elsewhere, according to the health services agency.

Chelsea's received from facility staff dealing with financial responsibility say it is those who are vulnerable have been out of money for healthcare their life, not who are suffering from a lack of access. "You might think after you tell that provider is not available, you should give the load to somebody in Russia or Sweden," she

Figure 3: **Unconditionally generated text of RADD-SE.**

Hi, my name is Shade-Rayhelynis-Neelsons. Interviewer: Two days ago, so let me speak to you in brief.Drake: Hi, are you studying for graduate school in late July, and somebody is interviewing you for you for his classes. My first personal quote is: so when they're doing class they're going to a shower, and they expect me to not be part of the shower any way. I have a take, I want to check when something's not right and make sure I spot it so that someone can get it happening. My hair is an important piece of me, and I can be the one complete human being that when I have a shower and say, "Okay, I really like my hair, and this is the thing that I would like and I want to believe something, I should just have an attitude check." Do you think anyone else can have time with me? Do I say I don't?I mean I get to 7 right the time I go to class, I sit back there and I feel like no one knows what to do. So, in my case I am not anxious, I'm just saying, "I feel like I have my hair without letting go from day to day, I've got to feel like I've been like the thing they should all be excited by."So, the days follow me, you start looking at my pictures, and you realize how pretty you are. It was just so big this moment for that office, because that was seeing positive things.I'm starting to grow up and be beautiful too.I mean–I mean it's kind of fun, now seeing beautiful girls, especially really in their 20s, come from unusual backgrounds. Back in the 50s I met people at one of the first places Woosz made mons, asked us to visit charity walks, and he would buy us a suit. And actually he really wanted to, so I know what the inspiration is. It's something that someone could have found out, and where they're from – I think it is like everyone is finding ways to relate.This is one of those things I remember the most about when I went to order the products!For example, it may be taken after the sporting event, and I don't know, but they asked us to choose color for our favorite parts in the group....they asked us to take their favorite color, then they made their eyes for each skin.The one that was the round head, like that one I chose so much, but it was really a really painful transition...and I don't know what touched my skin, so I don't know what I would do with it, I will tell you...that was not what I focused on, but I think I may have pretty much my original o-still darker hair, so I chose to go with the round head which I did really. Anyway, my job was to get the hair done, and I do most of my hair for school because I got married and had so many children while I was So. So, I'm always thinking again not to be confused. I had just been doing niggly since I was a child, and there was no reason to do it like that. The career had got started, and therefore I was really going all over the world no less. So one of the things I did was taking of his shirts so I look at him all in one go, I surprise him.I've got some of the most cool things going with those two of this. You see these all look alike, clean, and awesome–and they used to have a shop in the....a shop like that, they never sells coats, so that was really helpful to those guys, I feel, guys going into shop with these guys, and those guys can have the right color, they can have it that right look.I love my hair, it's like when I was a year old when I see scent in your nursery, to get the grandest response, I keep using the smell. I bring it to its level, honestly, it makes the weight apply.rake: Probably my favorite word to explain, after you explain, it isn't hard–in case that's your thing, I'm already who kind of like going, and I'm able to get access when I'm trying. I've been trying to finish coloring for people, for years, so I would like to work on it's own but the colors there to do color pattern are picked for another reason to put the final color off, that's like the other one for giving glitter or outline.So I'm just going to be looking side by side at coloring over and over again. It's my vision. Each color is my dream.As you know, a color is just simply something tucked into a dye and what makes perfect or tail end to a hair is...and that is why I always shampoo twice a day and shower three times a day.

Figure 4: **Conditionally generated text of RADD-SE.** Prompt tokens are highlighted in blue.

as well. However, she did not sign.

Gov. Johnluaj said the Amal effectively took the case to the Supreme Court.

"I was wrong to say that they left the Constitution in place and this is basically unconstitutional," Jackson said. "I don't think they're saying that. I'm. That means on the one hand they're going to have to intervene, or on the side of the other, they'll have to intervene. I can't think of changing a constitutional decision with them."

When Gov. Barack Obama announced Tuesday — and a federal court is set to challenge the way residents of the states violated the law — Attorney General Eric Holder chose to hold his own hearing. The entire state had a deadline to field a recipient of the letter asking for comment.

Because hearings were held before, Ohio and 17 states have each had such a case before.

At last night's hearings, more than four separate arguments were heard by a 52-45 margin in order to pass the repeal bill by a vote of 63-2 49 to 45. The bill also included Obamacare legislation and was pushed through Congress after opposition from 24 states. Both brought in a new governor, popular Gov Sen. Phil Bryant, another Republican in the Senate. Neil LePage neared an attorney in both cases and refused to find a new insurance secretary.

Both House Leader Mitch McConnell and Republicans said they would repeal the law entirely. The law would have been in the Oval Office of the Logged since 2011.

Former Judge Anthony Teague, the Ohio Chief Judge, found the overwhelming majority vote in favor of the legislation well in line and said it was a "needed forward."

"The things citizenship issues should go from statutes states have to regulations," he said. "They've got this idea that the courts are getting knocked to their own corner. And once they see it turn around them, sure as hell they'll have faith dogged by judges exercising constitutional rights."

Attorney Holder, the assistant secretary of state for policy at the Department of Justice, has discussed the idea that federal courts such as the U.S. Supreme Court should handling legal issues such as making tough immigration decisions for illegal immigrants.

"The thought process of helping illegal immigrants goes beyond the judicial process. What I understand...criminal immigration measures, gang activity, affirmative action efforts, criminal status," Holder said. "And things like that, we expect in the state to go to a long way. There clearly needs to be a criminal justice program on immigration and reform, and we need to recognize those efforts not to start."

Holder has also said it is a "real issue" for the constitution if the letter is signed off. He said it was essentially a message, seeking to show the majority "power of respect and the power of institutions."

But the attorney general said he hoped it would be a task to figure out how to start safeguarding each of their citizens' rights while restoring their constitutional trust.

"The only thing I think we can really do is have the judges to understand the nature of the judiciary, how powerful it is to be involved and to interfere with the government with no accountability on what path they want to move down," he said.<|endoftext|>Ex76561 molds at the worst parts of the UK economy will tell in the future – most poorer areas of England have suffered more than any person during the first years of 2008, according to ex-Home Secretary Jeremy Foot.

A total of 23,000 people of disposable incomes who have five mortgages – more than 4,000 households – will be considered as home buyers, even though published figures will be different from September, say researchers

Residents of more than 3,000 homes will be the third most likely to die because of jobs which fell in average terms home ownership during 2008 until the end of this year.

It's a surprising drop, according to the Wall Street Journal, which says the economy and the top 1

Professor Jeremy Foot, Home Secretary and the Information Society, raised concerns about the collapse of the UK's housing market – down from 39

He said: "The measure of who 'invested' at that time, doesn't include the number of people or businesses with the assets. There were only three big cities, New York were the other three?

"This year – it was announced that Royal Bank of Scotland would be the first to close in 20 years – it turned out then that the poorer areas, including by 2009 and 2010, were hammered hardest," Professor Foot said.

Professor Foot's lecture, which was released today in

Figure 5: **Unconditionally generated text of RADD-CE.**

I have a ick of death — that's where I am. That's what I need. That need to take something else. But I don't have that in my family. And here's why you might let it go. You don't want to really say what I wrote in my story, open your mind and finish it with purposeful thought. But I can do it if I get cancer."

"I can not help but accept that you brother-in-law was really on me and that that's not how I need to be," I said with my father's sad smile. He let it go a few days later, but it didn't prevent him from thinking about it. But I made sure he wouldn't let-in-law leave.

My mother saved my only father for the life. And he was a falsehood, but no doubt. He saved me.

"My mother thank god. But do you ask outside of me the questions, ask him. He's dealing with this and prepare for their perception of bad things he said. They should't forget him because the bad thoughts come with it."

I blackened my father and watched them walk back up to him in front of a good news line. He stood with us for two days; we stared on from spring to spring.

"It's OK." He said.

"I know what you want when you come here."

"I'll accept that," he said. "What I thought to look back was bad. I'm going to let it go and make an exception in this case. I came here after sharing the story. I've been looking to the world. And I don't care for anyone in this world, but I want to make it harder for other people to try to make this mistake."

He didn't justify what he meant. My mother drove us the way I trucked him back to his house, and he chuckled at me in simple words:

"I said to him I need to still be here, but do you want me to have this for the whole day?"

"These are my expectations. If we smile and I'm laughing thank you can I be so happy?"

Outside of my family, there was joy, joy, sorrow. At youngest, most of all, my father also saw the world. He suffered from hunger, his family lost access to wheelchair and every room. They fought with him sometimes, and when they needed to rest, he stepped back. I can only imagine, deep down, his family grew enormously with him.

Our journey with my parents at times had been a combination of things. We were all in pairs, and my brothers were such. Repeatedly we asked them to tell me when we wouldn't last to get that end. And I started missing my trip — never visited neighbors before. And finally one day I lost my patience. I all wanted to wander down to my father's house in76561. On my own, I couldn't see my beautiful mother again. I had an ear tumor that was brainblown away and lost all the ways to make the most of the time again. I took the bed from the entire bear family along with his three children, and fixed him up on one side and led him home through the open of the front door with my honey brother. We were hearing all of the other things I had heard were outlandish stories. Finally, he remained so grateful for his admiration for my dad.

What was possible my father really didn't have?

On November 2, 2013, my mother still did not show any empathy for my father. The loyalty he had was the only expression he had.

On the night where he was arrested, other members of his family noticed that they couldn't bribe a co-worker to walk up from his job. That led to him thinking, "I want you to not be a badexample in your family. You can't serve an individual who will make up his stupid demands for you in order to make a profit." For me, to fulfill the compassion of my own brother, I've become an able example of that, that kids.

With this decision, it was freeing that I had allowed him to try to decide how to make a better life for his family. I had allowed the animals to have a day off. This is something he and I can do. His son had thought they could, and it was relief he had been rearranging that reality. The time our family spends on the house was at seven times a day.

Figure 6: **Conditionally generated text of RADD-CE.** Prompt tokens are highlighted in blue.

599 # H  License

600 URL and license for existing assets we used are provided in Table 4.

Table 4: URL and license for existing assets we used.

| Name | URL | License |
|------|-----|---------|
| SEDD | `https://github.com/louaaron/Score-Entropy-Discrete-Diffusion` | MIT License |

