# OpenReview forum: "Your Absorbing Discrete Diffusion Secretly Models the Conditional Distributions of Clean Data"
_NeurIPS.cc/2024/Conference — Submitted to NeurIPS 2024_

### Official Review · Reviewer_vbN2 · 2024-07-12

**Soundness:** 3
**Presentation:** 3
**Contribution:** 3
**Rating:** 6
**Confidence:** 4

**Summary:**

This paper proves that the probability ratio that appears when computing the time reverse rate matrix for an absorbing state diffusion model has a simple form composed of the conditional distributions of clean data given partial masking scaled by an analytic time dependent weighting. They exploit this form to simplify the parametrization of absorbing state diffusion models and show this improves performance and sampling speed on text datasets.

**Strengths:**

This work is clearly written and I think theorem 1 will be genuinely useful for future work in absorbing state diffusion models. The fact that the time reversal of the rate matrix has a simple relation to the conditional distributions of clean data that is independent of time makes the target of optimization much clearer. This removes needless complexity when trying to condition models on time, even though the relationship with time is known analytically. This also helps model convergence since the scale factor of the target is known allowing the network to be targeting normalized quantities which is highly desirable for neural net training.

The removal of the time conditioning also has a significant benefit with respect to model speed ups. It is quite surprising that the absorbing state literature does not use this trick where at most L neural network evaluations are required for L length data. This paper should significantly help existing implementations in this regard by removing needless calls to the network.

**Weaknesses:**

Theorem 2 is wrong. Line C.14 in the proof is incorrect, it's not an equality but a lower bound. The correct version should read

$q\_\theta(x\_0) = \sum\_{\pi} U(\pi) q\_\theta(x\_0 | \pi)$

$q\_\theta(x_0) = \sum\_{\pi} U(\pi) \prod\_{l=1}^d q\_\theta(x\_0^{\pi(l)} | x\_0^{\pi( <l )} )$

$q\_\theta(x_0) = \mathbb{E}\_{\pi \sim U(S\_d) } [ \prod\_{l=1}^d q\_\theta (x\_0^{\pi(l)} | x\_0^{\pi (<l)}) ] $

$ \log q\_\theta(x\_0) = \log ( \mathbb{E}\_{\pi \sim U(S\_d)} [ \prod\_{l=1}^d q\_\theta(x\_0^{\pi(l)} | x\_0^{\pi(<l)}) ] )$

$ \log q\_\theta(x\_0) \geq \mathbb{E}\_{\pi \sim U(S\_d)} [ \sum\_{l=1}^d \log q\_\theta (x\_0^{\pi(l)} | x\_0^{\pi(<l)} ) ]$

When your generative model is a mixture of different generation paths (which an absorbing state diffusion model is), then you need to apply Jensen's inequality. See https://arxiv.org/pdf/2110.02037 equation (2).

Therefore, the authors should remove Section 3.3. I think this section should be replaced with discussion of Autoregressive Diffusion Models https://arxiv.org/pdf/2110.02037 which the author's model has basically reduced to. Autoregressive diffusion models randomly sample a generation order and gradually infill tokens with no dependence on time. The link between Autoregressive Diffusion Models and absorbing state diffusion should be clearly discussed in this paper and this reference is glaringly missing.

The authors should also remove line SEDD-S* in Table 2 since it is based on Theorem 2. Your models are then really not doing favourably compared to standard SEDD. What is your explanation for this and new narrative for Table 2?

In the paper's current state I cannot recommend acceptance since a large part of the narrative is based around Theorem 2. However, I believe the contributions surrounding Theorem 1 with regards to making models simpler and achieve good speed up stand alone as a worthy contribution. Therefore, if the authors clearly describe how they will adjust the narrative under this new information I will be happy to raise my score.

I think it would also be good to include a baseline against autoregressive diffusion models since they propose additional tricks relating to picking how many tokens to reveal. However, I appreciate this would be difficult in the limited time of the rebuttal period and is not required for an increase in score.

**Questions:**

In Figure 2, why is RADD with cache able to achieve such lower perplexity compared to no cache? In the limit of many steps, shouldn't these methods be performing the same? It is then a bit suspicious that with the cache is performing so much better. I think this could have something to do with the fact that generative perplexity should also be given with entropy measurements of the samples e.g. Table 1 in https://arxiv.org/pdf/2211.15089. This is because some models can have low entropy and 'good' generative perplexity which could be happening with your model.

**Limitations:**

The authors adequately discuss the limitations in Section 6.

---

> ### Author Rebuttal · Authors · 2024-08-06
>
> # Response to Reviewer vbN2
> Thank you for your extremely thorough review and constructive feedback on our paper. Below, we address your concerns and suggestions.
>
> ### Correction of Theorem 2 and Corresponding Experiments
> - **Modification on Section 3.3**:  We acknowledge the error in Theorem 2. Eq.(C.14) in the proof is indeed a lower bound, as you correctly pointed out. To address this, we will remove the current Section 3.3 and replace it with a discussion on any-order auto-regressive models (AO-ARM) [1*,2*,3*] as suggested. Specifically, the revised section will discuss the equivalence between the absorbing discrete diffusion objective of DCE loss and any-order autoregressive training objective.
>     #### Revised Proof Structure
>     - **Appendix C.1 and C.2**: We will retain the first two steps  corresponding to Appendix C.1 and C.2, which prove the equivalence between the DCE loss and Eq.(C.13):
>         $$d \mathbb{E}_ {k \sim U(\{1, \cdots, d\})} \frac{1}{k} \mathbb{E}_ {\tilde{\boldsymbol{x}} \sim U\left(\tilde{\mathcal{X}}_  k\right)}\sum_ {\tilde{x}^i=[\mathbf{M}]}-\log q_\theta\left(x_0^i | \tilde{\boldsymbol{x}}^{\mathrm{UM}}\right).$$
>
>     - **Appendix C.3**: In the last step in Appendix C.3, we will remove the unnecessary parts and directly reduce Eq.(C.13) to Eq.(C.19), which is the training objective of any-order autoregressive models:
>         $$d \mathbb{E}_ {k \sim U(\{1, \cdots, d\})} \frac{1}{k} \mathbb{E}_ {\pi \sim U\left(S_d\right)} \sum_{r=d-k+1}^d \log q_\theta\left(x_0^{\pi(r)} \mid x_0^{\pi(<d-k+1)}\right).$$
>
> - **Modification of Table 2**:  We will make the corrections to Table 2 as suggested. Specifically:
>   - We will remove the SEDD-S* line from Table 2, as it is based on the incorrect Theorem 2.
>   -  After fixing the bug mentioned in our overall response, we will update the DSE and DCE results for RADD-small and RADD-medium in Table 2, which show that RADD models outperform SEDD models.
>   -  We will also add baseline results of any-order autoregressive models (AO-ARM) trained on Eq.(C.19).
>
> The complete revised table  will be as follows:
>
> **Table B**
>
> | Method             | LAMBADA   | WikiText2 | PTB        | WikiText103 | 1BW       |
> | ------------------ | --------- | --------- | ---------- | ----------- | --------- |
> | SEDD-small Absorb  | 50.92     | 41.84     | 114.24     | 40.62       | 79.29     |
> | RADD-small DSE     |  **49.57**     | 38.83    | 111.74     |  37.46      | **72.35**     |
> | RADD-small DCE     | 50.56     | 39.02     | **109.03** | 36.38       | 72.60 |
> | AO-ARM-small       | 50.27 | **38.26** | 110.38     | **35.90**   | 74.28     |
> | SEDD-medium Absorb | 42.77     | 31.04     | 87.12      | 29.98       | 61.19     |
> | RADD-medium DSE    | 42.30 | **29.17** | **75.16**  | **28.03**   | 57.45 |
> | RADD-medium DCE    | 43.24     | 30.19     | 78.77      | 29.36       | 57.95     |
> | AO-ARM-medium      | **41.96**          | 29.96          |   79.06         |          28.51   | **57.07**|
>
>
>
>
> ### Clarification on Figure 2 Perplexity with and without Cache
> We appreciate your insightful analysis and the questions raised regarding the performance discrepancy between the RADD model with and without cache in Figure 2. Based on your suggestion, we have conducted additional entropy measurements for both SEDD and RADD models. We replicate the sampling 1024 times independently and compute the mean/std of the unigram entropy. The results are as follows:
>
> **Table C**
>
> Unigram Entropy for SEDD-small model
>
> | Steps    | 32          | 64          | 128         | 256         | 512         | 1024        | 2048        | 4096        |
> | -------- | ----------- | ----------- | ----------- | ----------- | ----------- | ----------- | ----------- | ----------- |
> | Euler    | 8.19 ± 0.13 | 8.07 ± 0.16 | 7.97 ± 0.20 | 7.86 ± 0.23 | 7.73 ± 0.27 | 7.59 ± 0.33 | 7.44 ± 0.34 | 7.25 ± 0.38 |
> | T-$\tau$ | 8.17 ± 0.13 | 8.06 ± 0.16 | 7.96 ± 0.20  | 7.86 ± 0.20 | 7.74 ± 0.24 | 7.57 ± 0.31 | 7.42 ± 0.32 | 7.22 ± 0.43 |
>
> **Table D**
>
> Unigram Entropy for RADD-small-dce model
>
> | Steps | 32          | 64          | 128         | 256         | 512         | 1024        | 2048        | 4096        | 8192        | 16384       | 32768       | 65536       |
> | ----- | ----------- | ----------- | ----------- | ----------- | ----------- | ----------- | ----------- | ----------- | ----------- | ----------- | ----------- | ----------- |
> | E-NFE | 32          | 64          | 127.96      | 251.35      | 442.84      | 647.48      | 805.98      | 906.13      | 962.64      | 992.69      | 1008.18     | 1016.05     |
> | Euler | 8.20 ± 0.13 | 8.11 ± 0.15 | 8.00 ± 0.17 | 7.90 ± 0.21 | 7.78 ± 0.23 | 7.64 ± 0.31 | 7.49 ± 0.31 | 7.30 ± 0.33 | 7.10 ± 0.39 | 6.90 ± 0.31 | 6.68 ± 0.47 | 6.35 ± 0.71 |
>
> It shows that your analysis is correct. Both the SEDD-small and RADD-small-dce models exhibit a decrease in unigram entropy as the number of sampling steps increases. This decrease suggests that the models generate simpler sentences with more sampling steps, which corresponds to the observed lower perplexity scores. Therefore, the perplexity doesn't converge in the limit of many steps.
>
> The underlying cause of this phenomenon remains unclear and seems to be a common issue for masked language models. We will add the experiment and discussion in the revision. If you have further suggestions, we welcome your feedback.
>
>
> [1*] Benigno Uria, Iain Murray, and Hugo Larochelle. A deep and tractable density estimator. ICML, 2014
>
> [2*] Emiel Hoogeboom, Alexey A. Gritsenko, Jasmijn Bastings, Ben Poole, Rianne van den Berg, and Tim Salimans. Autoregressive diffusion models. ICLR, 2022.
>
> [3*] Andy Shih, Dorsa Sadigh, and Stefano Ermon. Training and inference on any-order autoregressive models the right way. ICML, 2022.

---

> > ### Comment · Reviewer_vbN2 · 2024-08-08
> > **Response to Authors**
> >
> > Thank you for the response, I appreciate the new narrative and removing Theorem 2. The new results with the fixed bug look good and is reassuring to see.
> >
> > I still have a question regarding Figure 2. I appreciate the investigation into the entropy and it would be good to include this analysis in the paper. But regarding the difference between RADD small (cache) and RADD small (no cache) in Figure 2, I'm still unsure why there is a difference. Have you tried sampling RADD small (no cache) with say 65k steps and does it align with RADD small (cache) at 1024?

---

> > > ### Author Response · Authors · 2024-08-09
> > > **Further Discussion**
> > >
> > > Thank you for your follow-up and for appreciating the changes. To directly address your concern: the perplexity (PPL) of RADD small (no cache) at NFE = 65536 steps is equivalent to that of RADD small (cache) at E-NFE = 1016.05 steps. It seems that the confusion regarding Figure 2 may have arisen from a lack of clarity in our experimental setup. Let me clarify a few key points:
> > >
> > > Firstly, there are three closely related concepts: the Expected Number of Function Evaluations (E-NFE), the Number of Function Evaluations (NFE), and the the number of sampling steps. When no caching is used, these three concepts are equivalent. However, when caching is enabled, NFE becomes a random variable, and E-NFE is used to represent the time cost in practice. Due to caching, E-NFE is less than the number of sampling steps.
> > >
> > > Secondly, regardless of whether caching is used, the perplexity at the same number of sampling steps remains identical, similar to how KV cache works. As a result, We only conducted one set of experiments with caching, varying the number of sampling steps, and plotted the green line using E-NFE on the x-axis. For the yellow line representing the no-cache scenario, We plotted NFE directly as the x-axis.
> > >
> > > Regarding your specific question about running a 65k-step sampling evaluation without caching, it would indeed take several days to complete. However, since the perplexity remains the same with or without caching at the same number of sampling steps, we can compare results directly using the green line (with cache) to infer the expected performance at higher steps. According to the relationship between steps and E-NFE shown in Table D, the perplexity of RADD small (no cache) at NFE = 65536 steps is equivalent to that of RADD small (cache) at E-NFE = 1016.05 steps.
> > >
> > > We hope this explanation clarifies the rationale behind the plots. Reviewer 9MAq also raised similar concerns, so you may refer to our response to them for additional context. We ensure to clarify these points in the revised manuscript. Please let us know if further details are needed or if you have any other concerns.

---

> > > > ### Comment · Reviewer_vbN2 · 2024-08-09
> > > > **Response to Authors**
> > > >
> > > > Thank you for the clarifying comments, I understand better now the consistency of Figure 2. I will raise my score to 6.

---

> > > > > ### Author Response · Authors · 2024-08-10
> > > > >
> > > > > We're glad our response was helpful and appreciate the positive score adjustment. We'll ensure the final version reflects the promised revisions. Thank you again for your thorough review and valuable feedback.

---

### Official Review · Reviewer_9MAq · 2024-07-12

**Soundness:** 3
**Presentation:** 3
**Contribution:** 3
**Rating:** 6
**Confidence:** 4

**Summary:**

This paper proposes a simplified discrete diffusion model to improve upon prior language diffusion models.

**Strengths:**

* The method is simple and scalable. It is overall a nice insight, and the authors do a good job in extracting the relevant and impactful applications of this.

* The method seems to improve upon previous results, in particular resulting in better/faster sample quality.

* The presentation is pretty clear and direct.

**Weaknesses:**

* Although the method speeds up sampling, especially in the large sample step regime, this is a bit misleading/irrelevant. In particular, under more standard sampling practices, the gain is naturally not as big, so the claim of 3.5x improvement is a bit misleading. Furthermore, this does not really improve the sample quality at a smaller number of steps, which is the critical question. As a comparison, this would be like sampling from a standard diffusion model with 4096 timesteps, showing that you can speed it up in that regime, and then claiming a general improvement.

* The results are ultimately a bit marginal. The improvements on sample quality are nice, but I think there is a mistranslation between figure 2 and table 1 (there is no 15 generative perplexity for RADD in that table). Until this is clarified, I'm trusting the results of table 1 more. Table 2 also shows a slight improvement.

* The exact likelihood computation is never applied. I want table 2 to showcase this exact likelihood instead of just a bound.

* The model size is only small. I want to see a similar improvement for the medium quality.

**Questions:**

N/A

**Limitations:**

Yes

---

> ### Author Rebuttal · Authors · 2024-08-06
>
> # Response to Reviewer 9MAq
> Thank you for the detailed review and acknowledgment of our contributions.  Below we address specific questions.
> - **Q1: Misleading speeds up declaration.**
> - A1: We apologize for any confusion caused by the initial claim of our speed-up results. We will revise our statement to emphasize that the speed-up is significant mainly in large steps (e.g., 4096), and the 3.5x speed-up does not generalize to all steps.
>
>
> - **Q2: Mistranslation between Figure 2 and Table 1**
> - A2: Sorry for the confusion regarding the discrepancy between Figure 2 and Table 1. The yellow line and the green line in Figure 2 represent the same perplexity results but with different x-axis.
>   Specifically:
>   - **RADD Small without cache (yellow line)**, the x-axis represents **NFE**, equivalent to sampling steps in this context.
>   - **RADD Small with cache (green line)**, the x-axis represents **E-NFE**, which is less than the actual sampling steps due to the efficiency introduced by caching.
>
>   For space constraints in the figure, we presented 8 data points for RADD Small without cache and 11 data points for RADD Small with cache. However, only 8 data points are shown in Table 1 to for consistency in the comparison. In the revised version, we will update the x-axis label in Figure 2 to 'NFE/E-NFE' and provide more detailed clarifications.
> - **Q3:The exact likelihood computation is never applied. I want table 2 to showcase this exact likelihood instead of just a bound.**
> - A3: We appreciate your feedback on the exact likelihood computation. Another reviewer pointed out an error in our proof in Theorem 2, which impacts the exact likelihood computation. As part of our response to Reviewer vbN2, we have acknowledged the error in Theorem 2 and committed to revising. Please refer to the first section of our response to  Reviewer vbN2 for detailed corrections and updated experimental results.
> - **Q4: The improvements are marginal. The model size is only small. I want to see a similar improvement for the medium quality.**
> - A4: We have updated our results to demonstrate the effectiveness of RADD for both small and medium models. Please see the overall response section for updated results. It shows that our methods also apply to medium-sized models with consistent improvements over SEDD in all zero-shot tasks.

---

### Official Review · Reviewer_7sHz · 2024-07-12

**Soundness:** 2
**Presentation:** 3
**Contribution:** 3
**Rating:** 6
**Confidence:** 2

**Summary:**

This work derives a new interesting connection between the concrete score and conditional target densities in absorbing diffusion models, which decomposes the time-dependent ratio between marginal probabilities (of two transitive states) as a conditional distribution on clean data scaled by an analytic time-dependent scalar, and hence inspires the commonly-used scaling trick and new re-parameterizations. In addition, it also simplifies the original complicated loss objective (denoising score entropy; DSE) as a more straightforward denoising cross-entropy loss (DCE) that enables the exact log-likelihood computation.

**Strengths:**

1. This paper is generally well-written and easy to follow.
2. This work proposes valuable insights of decoupled model parameterizations and simplified learning objectives, whose effectivenesses are theoretically grounded.
3. The proposed methods are also numerically verified, which advances the development of (absorbing) discrete diffusion models.

**Weaknesses:**

1. Despite that the overall presentation is good, it can be further improved by interpreting or illustrating more about the problem formulation. For example, what is the intuition behind the absorbing matrix $Q^{\text{absorb}}$ (eq. (2.4))? Why do we require a more complicated DSE loss instead of usual score-matching objectives (e.g. MSE)? Note that in eq. (2.6), the score network must be *additionally* positive.
2. Although the proposed method (RADD) is reported to be superior for efficient sampling, the performance of RADD need further verifications on language modeling tasks (Table 2). The hyper-parameters should be fine-tuned to better demonstrate the capability of RADD.

**Questions:**

1. Please provide more details of questions raised in the weaknesses section above.
2. Detail 1: Why does Kolmogorov’s forward equation (eq. (2.3)) hold in this time-dependent setting? Note that the studied continuous-time Markov chain with discrete states ($Q$-process) is time-inhomogenous, since the transition rate matrix $Q_t$ is dependent of $t$.
  - When the Markov chain is time-homogenous, $Q_t\equiv Q$, and Kolmogorov’s forward equation follows from Chapman–Kolmogorov
equation (whose derivation requires the time-homogeneity).
3. Detail 2: What is the intuition to connect the form of $Q^{\text{absorb}}$ (eq. (2.4)) with the absorbing process? Are there any other forms that also work in practice?

**Limitations:**

As is stated by authors, future explorations include flexible variable-length texts generation and applications to models with larger scales.

---

> ### Author Rebuttal · Authors · 2024-08-06
>
> # Response to Reviewer 7sHz
>
> Thank you for acknowledging our contributions. We have tailored our rebuttal to address the points you raised.
>
> ### Weaknesses:
>
> 1. **Problem Formulation:**
>     We will add more intuitive explanations and illustrations regarding the problem formulation. Below are the clarifications:
>    - **Intuition Behind the Absorbing Matrix $\mathbf{Q}^{\text{absorb}}$ (eq. (2.4)):**
>    In continuous-time Markov chains, the transition rate matrix $\mathbf{Q}_t = \sigma(t)\mathbf{Q}$ indicates the probabilities of moving from one state to another over an infinitesimally small time interval. For $\mathbf{Q} =\mathbf{Q}^{\text{absorb}}$, each state $i$ has a transition rate of 1 to the absorbing state (last state) and a rate of -1 on the diagonal representing leaving the original state. This ensures that once the system reaches the absorbing state, it remains there permanently.
>    - **Reason for DSE Loss Instead of Usual Score-Matching Objectives (e.g., MSE):** MSE loss is derived under the assumption of Gaussian noise, which is not suitable for discrete data. It cannot guarantee that the concrete score remains positive, leading to poor performance. For more details, you can refer to Sections 2.2 and 3.1 in SEDD[1*].
>
> 2. **Performance Verification of RADD:**
>    - We have updated the Table 2 to better demonstrate the effectiveness of RADD. See Table A in the overall response section for the updated results. After fixing the bug regarding to the ignored layernorm bias and hyperparameter tuning, it shows that two RADD models consistently outperform SEDD in all zero shot tasks for both model size.
> ### Questions:
>
> 1. **Kolmogorov's Forward Equation in Time-Dependent Settings:**
>    - Kolmogorov's forward equation holds generally for time-dependent settings. Its derivation is based on the Total Probability Theorem, making it applicable to both time-homogeneous and time-inhomogeneous Markov processes. For a detailed explanation, please refer to Section 4 in Feller [2*].
>
> 2. **Intuition and alternative forms of  $\mathbf{Q}$ :**
>    - The form of $Q^{\text{absorb}}$ is chosen to model states that transition into absorbing states where no further transitions occur. To the best of our knowledge, only the absorbing form and uniform form have been used in practice, with the absorbing form performing the best. Other forms may be inefficient in terms of computation and memory, as discussed in Section 3.3 of SEDD[1*].
>
> [1*] Aaron Lou, Chenlin Meng, and Stefano Ermon. Discrete diffusion modeling by estimating the ratios of the data distribution, 2024
>
> [2*] W Feller. On the theory of stochastic processes, with particular reference to applications. In Proceedings of the [First] Berkeley Symposium on Mathematical Statistics and Probability. The Regents of the University of California, 1949.

---

> > ### Comment · Reviewer_7sHz · 2024-08-11
> >
> > Thanks for the reply! I have no further questions and will raise my score to 6.

---

> > > ### Author Response · Authors · 2024-08-12
> > >
> > > Thank you for your positive feedback. We will make the necessary revisions in the final version as promised.

---

### Author Rebuttal · Authors · 2024-08-06

# Overall Response


We would like to thank all the reviewers for taking their time to review our paper and provide high quality feedback. We have updated the results of RADD to better demonstrate the effectiveness of RADD, addressing the common concerns from Reviewer 7sHz,9MAq, and vbN2.

## Updated Zero-Shot Language Modeling Perplexity Results

In the initial version of the RADD model, we removed the time-dependent adaptive layer normalization but forgot to add a bias term to the time-independent layer normalization.  After correcting this bug, we retrained the RADD model and updated the results for both the small and medium RADD models, as shown in the table below. The best results for each model size are highlighted in bold.

**Table A**

| Method             | LAMBADA   | WikiText2 | PTB        | WikiText103 | 1BW       |
| ------------------ | --------- | --------- | ---------- | ----------- | --------- |
| SEDD-small Absorb  | 50.92     | 41.84     | 114.24     | 40.62       | 79.29     |
| RADD-small DSE     |  **49.57**     | **38.83**    | 111.74     |  37.46      | **72.35**     |
| RADD-small DCE     | 50.56 | 39.02 | **109.03** | **36.38**   | 72.60 |
| SEDD-medium Absorb | 42.77     | 31.04     | 87.12      | 29.98       | 61.19     |
| RADD-medium DSE    | **42.30** | **29.17** | **75.16**  | **28.03**   | **57.45** |
| RADD-medium DCE    | 43.24     | 30.19     | 78.77      | 29.36       | 57.95     |



Here, the SEDD Absorb models represent the strongest baseline models, trained using the scaling trick on DSE loss. For all RADD models, we set the dropout rate to 0.02 and the weight decay to 0.03, while keeping all other hyperparameters unchanged from the SEDD settings.

---

### Decision · Program_Chairs · 2024-09-25

**Decision:**

Reject

**Comment:**

This work observed the relationship between the discrete score model and the conditional mean of clean data in absorbing discrete diffusion models. Notably, it shows that the conditional mean of clean data is independent of time. These observations are then used to simplify the score entropy loss and accelerate sampling. Reviewer vbN2 pointed out an incorrect claim in the original submission, where the RADD loss was mistakenly identified as the log exact likelihood. During the rebuttal, the authors acknowledged this mistake and agreed to remove the section that repeats findings from the "Autoregressive Diffusion Models" paper.

Comment by SAC: While the authors did respond appropriately to characterize how to modify the paper, the consequence of this error and removal is that the paper's framing are affected throughout including in the first contribution, and segments of the methodology. This left the paper needing a large enough revision, leaving it ultimately a borderline paper which fell on the side of rejection. Once the changes herein (suggested by the reviewers) are incorporated, I encourage the author to resubmit their work with the updated framing.